# *Drosophila* PDGF/VEGF signaling from muscles to hepatocyte-like cells protects against obesity

Arpan C Ghosh[1]*, Sudhir Gopal Tattikota[1†], Yifang Liu[1†], Aram Comjean[1], Yanhui Hu[1], Victor Barrera[2], Shannan J Ho Sui[2], Norbert Perrimon[1,3]*

[1]Department of Genetics, Blavatnik Institute, Harvard Medical School, Boston, United States; [2]Harvard Chan Bioinformatics Core, Harvard T.H. Chan School of Public Health, Boston, United States; [3]Howard Hughes Medical Institute, Boston, United States

**Abstract** PDGF/VEGF ligands regulate a plethora of biological processes in multicellular organisms via autocrine, paracrine, and endocrine mechanisms. We investigated organ-specific metabolic roles of *Drosophila* PDGF/VEGF-like factors (Pvfs). We combine genetic approaches and single-nuclei sequencing to demonstrate that muscle-derived Pvf1 signals to the *Drosophila* hepatocyte-like cells/oenocytes to suppress lipid synthesis by activating the Pi3K/Akt1/TOR signaling cascade in the oenocytes. Functionally, this signaling axis regulates expansion of adipose tissue lipid stores in newly eclosed flies. Flies emerge after pupation with limited adipose tissue lipid stores and lipid level is progressively accumulated via lipid synthesis. We find that adult muscle-specific expression of *pvf1* increases rapidly during this stage and that muscle-to-oenocyte Pvf1 signaling inhibits expansion of adipose tissue lipid stores as the process reaches completion. Our findings provide the first evidence in a metazoan of a PDGF/VEGF ligand acting as a myokine that regulates systemic lipid homeostasis by activating TOR in hepatocyte-like cells.

**\*For correspondence:**
arpan_ghosh@hms.harvard.edu (ACG);
perrimon@genetics.med.harvard.edu (NP)

[†]These authors contributed equally to this work

**Competing interests:** The authors declare that no competing interests exist.

## Introduction

Specialized organ systems compartmentalize core metabolic responses such as nutrient uptake, nutrient storage, feeding behavior, and locomotion in multicellular organisms. In order to link systemic metabolic status to appropriate physiological responses, information on local metabolic events within each of these organs must be integrated. Inter-organ communication factors play an important role in mediating this process of systemic metabolic integration. For instance, classical metabolic hormones such as insulin and glucagon released from the pancreas, as well as leptin released from the adipose tissue, can act on multiple peripheral and central nervous system targets. While insulin and glucagon define anabolic and catabolic states of an animal, leptin limits food intake in response to adequate energy stores in the adipose tissue (*Ahima et al., 1996*; *Moore and Cooper, 1991*; *Schade et al., 1979*; *Tartaglia et al., 1995*). In addition to these classical hormones, a number of peptides have been identified that can mediate inter-organ communication axes. For example, adiponectin, adipisin, and asprosin are molecules that are released from the adipose tissue that signal to distant tissues such as muscle/liver and pancreas (*Lo et al., 2014*; *Romere et al., 2016*; *Yamauchi et al., 2014*). IGF1, angiotensin, and IGFBPs are released from the liver and signal to multiple distant organs including adipose tissue, muscle, and kidney (*Boucher et al., 2012*; *Clemmons, 2007*; *Droujinine and Perrimon, 2016*). The role of the skeletal muscle as an endocrine organ has gained special interest lately, primarily due to the beneficial effects of having healthy active muscles toward ameliorating or preventing the pathophysiology of a number of disorders and diseases (*Benatti and Pedersen, 2015*; *Boström et al., 2012*; *So et al., 2014*). A number of skeletal

muscle-derived signaling factors ('myokines'), including Irisin, Myostatin, IL6, Myonectin, and FGF21, have recently been characterized and they signal both locally, and to distant tissues, to control processes as diverse as muscle growth and browning of fat (*Boström et al., 2012*; *Fisher and Maratos-Flier, 2016*; *McPherron, 2010*; *Pedersen and Febbraio, 2012*; *Seldin et al., 2012*). Nevertheless, conservative estimates, based on bioinformatic analysis of the skeletal muscle transcriptome and proteomics studies, indicate that skeletal muscles are capable of secreting more than 200 myokines and the vast majority of these proteins are yet to be characterized (*Pedersen and Febbraio, 2012*). The vertebrate PDGF/VEGF signaling ligands are also secreted from the skeletal muscles; however, the biological roles of muscle-derived PDGF/VEGF ligands remain unknown (*Catoire et al., 2014*; *Henningsen et al., 2010*; *Raschke et al., 2013*).

The PDGF/VEGF family of signaling ligands have co-evolved with multicellularity and are proposed to signal in the context of organisms with tissue-level organization (*Holmes and Zachary, 2005*). VEGF family ligands, including VEGF-A and VEGF-B, influence metabolic responses such as adiposity, insulin resistance and browning of fat (*Elias et al., 2012*; *Hagberg et al., 2012*; *Lu et al., 2012*; *Robciuc et al., 2016*; *Sun et al., 2012*; *Sung et al., 2013*; *Wu et al., 2014*). However, mechanisms by which these molecules regulate lipid metabolism are complex and multiple context dependent and confounding models of their action have been proposed (*Elias et al., 2012*; *Hagberg et al., 2012*; *Lu et al., 2012*; *Robciuc et al., 2016*; *Sun et al., 2012*; *Sung et al., 2013*; *Wu et al., 2014*). Nevertheless, in all these studies the effect of VEGFs on obesity and insulin resistance have been attributed to their roles in modulating tissue micro-vascularization and endothelial cell biology. Similarly, PDGF signaling ligands PDGF-BB and -CC have also been implicated in adipose tissue expansion, glucose metabolism, and thermogenesis in beige fat, by influencing remodeling of tissue vascularization (*Onogi et al., 2017*; *Seki et al., 2016*). VEGFs and PDGFs are well-known regulators of vascularization and endothelial cell biology, and tissue vascularization definitely plays an important role in adipose tissue health, inflammation, and ultimately insulin resistance. Nevertheless, signaling to non-endothelial cell types by PDGF/VEGF ligands could play equally important roles in regulating obesity and insulin resistance.

In *Drosophila*, the PDGF/VEGF pathway ligands are encoded by three genes, *pvf1*, *pvf2*, and *pvf3* (*Cho et al., 2002*; *Duchek et al., 2001*). Similar to vertebrate PDGFs/VEGFs, these molecules signal through a receptor-tyrosine Kkinase (RTK) encoded by *pvr*. Once bound to the receptor, Pvfs are known to primarily activate the Ras/Raf/ERK intracellular cascade (*Duchek et al., 2001*; *Heino et al., 2001*). Phylogenetic analysis of the Pvfs show that Pvf1 is closely related to both VEGFs and PDGFs and most likely plays the dual role of representing both these divergent signaling pathways in the fly (*Holmes and Zachary, 2005*). Pvf2 and Pvf3 are more ancestral forms of the protein and are phylogenetically distinct with Pvf2 lacking the conserved cysteine necessary for forming the cysteine knot structure characteristic of this family of ligands (*Holmes and Zachary, 2005*; *Kasap, 2006*).

Core metabolic organs and signaling mechanisms that regulate metabolic homeostasis are also highly conserved between *Drosophila* and vertebrates (*Droujinine and Perrimon, 2016*). Metabolic organ systems that are functionally and structurally analogous to the vertebrate adipose tissue, muscle, intestine, and liver exist in *Drosophila* (*Gutierrez et al., 2007*; *Leopold and Perrimon, 2007*). The *Drosophila* adipose tissue (also called the fat body) functions as the primary site for lipid storage. The adipose tissue is also the key site for sensing the nutrient status of the animal and coupling it to systemic growth, metabolism, and feeding behavior (*Colombani et al., 2003*; *Grönke et al., 2007*). The roles of the liver are shared between the adipose tissue and specialized hepatocyte-like cells called the oenocytes, with the adipose tissue being the primary site of glycogen storage and the oenocytes playing the roles of lipid mobilization and synthesis of specialized lipid molecules (*Gutierrez et al., 2007*; *Makki et al., 2014*; *Storelli et al., 2019*). However, much remains to be learned about the biological roles of the specialized organ systems in *Drosophila* and how they relate to organ systems in vertebrates.

A number of hormonal signals emanating from the *Drosophila* adipose tissue, gut, and muscle have been characterized and shown to play roles as diverse as regulation of insulin and glucagon (AKH in *Drosophila*) release, nutrient uptake in the gut, and mitochondrial metabolism (*Chng et al., 2014*; *Droujinine and Perrimon, 2016*; *Ghosh and O'Connor, 2014*; *Rajan and Perrimon, 2012*; *Song et al., 2017*). Components of core metabolic pathways such as the TOR signaling pathway are also highly conserved in the fly and much has been learned about the fundamental principles of the

regulation of TOR and its roles in growth and aging from studies in *Drosophila* (*Antikainen et al., 2017*; *Colombani et al., 2003*; *Kim et al., 2008*; *Piper and Partridge, 2018*). *Drosophila* Pvfs have been primarily studied in the context of their roles in embryonic development, cell motility, and specification of immune cells (*Duchek et al., 2001*; *Ishimaru et al., 2004*; *Rosin et al., 2004*). However, the role of these signaling peptides in metabolism remains less explored.

Here, we identify the *Drosophila* PDGF/VEGF ortholog, Pvf1, as a tubular muscle derived signaling factor (myokine) that regulates systemic lipid stores by inhibiting lipid synthesis. Additionally, by subjecting the metabolically active organ systems that reside in *Drosophila* abdomen (muscle, oenocytes, and adipose tissue) to single-nuclei RNA sequencing (snRNA-Seq), we identify the adult oenocytes to be one of the primary targets of Pvf signaling. We provide further evidence that muscle-derived Pvf1 signals to the oenocytes where it activates the PvR/Pi3K/Akt1/TOR signaling pathway that in turn inhibits lipid synthesis in the organism. We also find that muscle-derived *pvf1* plays an important role in regulating the build up of adult adipose tissue lipid stores in newly eclosed flies. Flies eclose from their pupal case with limited adult adipose tissue lipid stores. Post eclosion they enter a developmental stage that is marked by increased lipid synthesis needed for accumulation of the adipose tissue lipid stores. *pvf1* expression in the muscle increases rapidly during this stage and premature expression of *pvf1* in the muscle interferes with the process of lipid accumulation. Taken together our study indicates that muscle-Pvf1 serves as an inhibitory signal for suppressing de-novo lipid synthesis once the adult adipose tissue has accumulated sufficient lipid stores at the end of the adipose tissue lipid build-up phase. Our snRNA-seq data of the adult abdominal region that includes adipose tissue, oenocytes, and abdominal muscles will also serve as an invaluable resource to further understand the biology of these organs. To facilitate visualization of this rich resource of gene expression profiles, we have created a searchable webtool where users can mine and explore the data (https://www.flyrnai.org/scRNA/abdomen/).

## Results

### Muscle-specific loss of Pvf1 leads to increased lipid accumulation in the adipose tissue and oenocytes

To investigate tissue-specific roles of Pvfs in regulating lipid homeostasis, we knocked down *pvf1*, *pvf2,* and *pvf3* in various metabolically active tissues in the adult male fly using temperature-inducible drivers and looked for effects on the adiposity of the fly. Strikingly, knockdown of *pvf1* in the adult muscle (*mus^ts^>pvf1-i*), but not in other organs such as the gut, adipose tissue, or oenocytes, was associated with a severe obesity phenotype characterized by increased lipid droplet size in the adipose tissue cells (*Figure 1A, B* and *Figure 1—figure supplement 1A*). Additionally, the experimental animals showed increased accumulation of lipid droplets in the oenocytes which are normally devoid of or have very few lipid droplets (*Figure 1A, B* and *Figure 1—figure supplement 1A*). *mus^ts^>pvf1-i* flies also showed significantly higher levels of whole animal triacylglycerol (TAG) content than control flies (*Figure 1C*). The increase in total TAG content was observed for two independent RNAi lines against *pvf1* (*Figure 1C*). The increase in TAG content of *mus^ts^>pvf1-i* flies was more pronounced when flies were challenged with a mildly high-sugar diet (15% w/v added sugar to our standard food) (*Figure 1—figure supplement 1B*). Note that we used this food condition for all our experiments unless mentioned otherwise. Knocking down either *pvf2* or *pvf3* in the adult muscle did not affect total TAG content of the animals, further demonstrating that the phenotype is specifically associated with muscle-specific loss of Pvf1 (*Figure 1D*). When fed with $^{14}$C-U-Glucose for 24 hrs, *mus^ts^>pvf1-i* and control flies showed comparable incorporation of $^{14}$C in whole fly homogenates (*Figure 1—figure supplement 1C*), indicating that *mus^ts^>pvf1-i* flies do not eat more than control animals and that the obesity phenotype observed is most likely caused by metabolic defects.

Since the density at which animals are reared can have a significant effect on metabolic parameters such as total protein and total TAG levels, we tightly controlled the rearing conditions for all our experiments (see Materials and methods for more details). We regularly retrieved ~ 30–40 males and an equal number of females from our crosses that were carried out in standard 10 cm food vials. We discarded vials that had an aberrantly low ($\leq$60) or high number ($\geq$80) of adults eclosing from them. The adult adipose tissue dissection, staining, and imaging protocol is quite laborious. This makes inclusion of a large number of controls in every experiment quite difficult. Therefore, we kept the

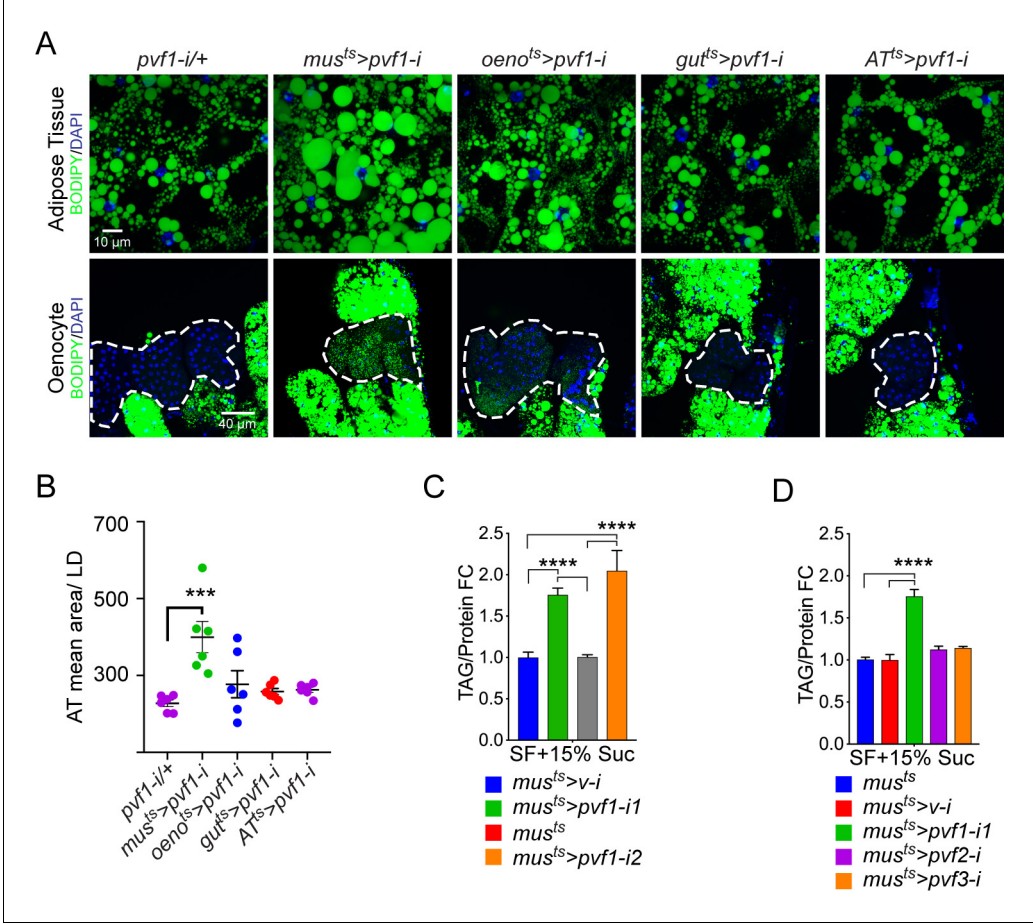

**Figure 1.** A muscle-to-oenocyte Pvf1 signaling axis protects against obesity. (**A**) BODIPY staining showing neutral lipid accumulation in the adult male adipose tissue (AT) and oenocytes (dorsal abdominal cuticle) of flies in which *pvf1* was knocked down using an RNAi transgene (VDRC: kk102699) specifically in the adult: muscle (*mus^ts^* = *muscle-Gal4^Gal80ts^*), oenocytes (*oeno^ts^* = *oenocyte-Gal4^Gal80ts^*), gut (*gut^ts^* = *gut-Gal4^Gal80ts^*), and adipose tissue (*AT^ts^* = *AdiposeTissue-Gal4^Gal80ts^*). The RNAi transgene alone is shown as a control. Similar results were observed using a different RNAi line (NIG: 7103 R-2, data not shown). (**B**) Mean lipid droplet size (≥10 microns in diameter) in the adipose tissue of flies shown in (**A**). Only muscle-specific loss of *pvf1* show a significant increase compared to the control genotype, N = 6 animals, One-way ANOVA followed by Tukey's HSD test, *** denotes p≤0.001, error bars = SEM. (**C**) Triacylglycerol (TAG) assay showing TAG/protein ratio in adult males in which *pvf1* was knocked down specifically in the muscle. Knocking down *pvf1* using two independent RNAi lines (*pvf1-i1*::VDRC kk102699 and *pvf1-i2*::NIG 7103 R-2) leads to a significant increase in total TAG content of the flies. (*mus^ts^>v-i* = *muscle-Gal4^Gal80ts^>vermilion* RNAi). (SF+15% Suc = standard lab food supplemented with 15% sucrose w/w). N = 6, Cohorts of eight males were used for each data point. One-way ANOVA followed by Tukey's HSD test, **** denotes p≤0.0001, error bars = SEM. (**D**) Total TAG content of adult males with adult muscle-specific (*mus^ts^*) knockdown of *pvf1*, *pvf2* and *pvf3*. N = 6, Cohorts of eight males were used for each data point. One-way ANOVA followed by Tukey's HSD test, **** denotes p≤0.0001, error bars = SEM.

The online version of this article includes the following figure supplement(s) for figure 1:

**Figure supplement 1.** A muscle to oenocyte Pvf1 signaling axis protects against obesity.

number of control genotypes that were processed in parallel in a given experiment to a manageable number while processing additional control genotypes separately to determine the baseline for lipid droplet size in the adult adipose tissue (*Figure 4—figure supplement 3A, B*). Overall, we looked at nine control genotypes (*pvf1-i/+* ǁ *oneo^ts^/+* ǁ *oeno^ts^>UAS* empty ǁ *inr^DN^/+* ǁ *tsc2-i/+* ǁ *akt1-i/+* ǁ *Pi3K21B-i/+* ǁ *Pi3K92E-i/+* and *UAS-tsc1, UAS-tsc2/+*) and the average lipid droplet area remained similar in all these controls. The use of GFP overexpression or knockdown of an unrelated gene such as vermillion is often used in the field as a genetic background control. These controls sometimes

tend to manifest unpredictable phenotypes either due to toxicity from GFP or yet unknown function for the presumed to be unrelated gene while providing little to no benefit in terms of being ideal genetic background control. Hence, we did not use them in our study. Rather we relied on the fact that multiple experimental approaches in our study culminated to the same logical end to draw our conclusions.

To verify the presence and distribution of Pvf1 protein in adult muscles, we immunostained the indirect flight muscles (IFMs) and leg muscles using an anti-Pvf1 antibody (*Duchek et al., 2001*; *Rosin et al., 2004*). Pvf1 is abundantly present in the striated tubular leg muscles (*Figure 1—figure supplement 1D*) where it is stored between the individual myofibrils (*Figure 1—figure supplement 1D1', D3'*) and is more concentrated at both the M and Z discs (*Figure 1—figure supplement 1D3,3',3'*). To verify the specificity of the signal, we immunostained the leg muscles of $mus^{ts}$>pvf1-i flies with anti-Pvf1 antibody and observed a strong reduction in Pvf1 protein level (*Figure 1—figure supplement 1D2',D4,D4'*). Interestingly, the IFMs did not show any staining for Pvf1 indicating that the protein is primarily stored in the striated tubular muscles in the fly.

## Single-nuclei RNA-sequencing (snRNA-Seq) identifies enrichment of PvR RTK signaling pathway in the Oenocytes

Single-nuclei sequencing presents unprecedented access to the transcriptomes of cell types residing in complex tissue structures or organs that are difficult to dissect and segregate (*Birnbaum, 2018*; *Chen et al., 2018*; *Kulkarni et al., 2019*; *Wu et al., 2019*). We took advantage of this tool to understand the transcriptomes of tissue types residing in the adult abdominal cuticle that harbors several metabolically active tissues such as the fat body, abdominal-muscles, and oenocytes, which are functionally analogous to adipose tissue, skeletal muscle, and liver, respectively, in higher vertebrates (*Droujinine and Perrimon, 2016*; *Musselman and Kühnlein, 2018*). To delineate the patterns of gene expression in each of these tissues, we dissected and dissociated a total of 80 adult fly abdominal cuticles (along with the attached adipose tissue and oenocytes) and subjected the single nuclei to 10X genomics-based (*Zheng et al., 2017*) single-nuclei RNA-sequencing (snRNA-seq) (*Figure 2A*). Two independent rounds of sequencing were performed on two biological replicates (with 40 animals per replicate) to obtain a median read depth of 8904 reads per nucleus (*Figure 2— figure supplement 1A*). Because tissue dissociations for the single nuclei preparations are often associated with the risk of ambient RNA contamination, our quality control pipeline included SoupX (*Young and Behjati, 2018*) to eliminate potential ambient RNA from our analysis. Further, we used Harmony (*Korsunsky et al., 2019*) that is integrated into Seurat (*Stuart et al., 2019*) to correct for batch effects in the two replicates to finally retain 15,280 nuclei with a median of 192 genes per nucleus for downstream analysis (*Figure 2—figure supplement 1B, D*; *Supplementary file 1*). Our clustering analysis at resolution 0.1 (see Materials and methods) identified 10 unique clusters, where three major clusters were assigned to adipose tissue, oenocytes, and muscle based on known markers for each of these tissue types including *apolpp*, *fasn3*, and *mhc*, respectively (*Figure 2B*; *Figure 2—figure supplement 1E*; *Supplementary file 2*). We validated our snRNA-seq data using GAL4 lines for certain top enriched and novel genes such as *Pellino* (*Pli*), *sallimus* (*sls*), and *geko* and found that they specifically express in adipose tissue, muscle, and oenocytes, respectively (*Figure 2— figure supplement 1F-I*). With regard to the rest of the minor clusters (4-10), we believe most of them are remnant tissues most likely pertaining to gut/malpighian tubule based on the enrichment of *alphaTry*, *Whe*, and *Mur18B* (Clusters 4–6, respectively; *Supplementary file 2*). On the other hand, we consider clusters 7–10 likely to be part of the ejaculatory bulb (Eb) as they are enriched in certain male-specific genes such as *bond*, *EbpIII*, *soti*, and *Ebp*, respectively (*Figure 2—figure supplement 1I-J*; *Supplementary file 2*). Interestingly, increasing the resolution of the clustering analysis to 0.4 leads to the identification of 15 unique clusters with unique gene expression signatures (*Figure 2—figure supplement 3*). Some of these clusters may reflect potential heterogeneity within the adult adipose tissue.

To explore the pathways that are enriched in each of these clusters, we performed pathway enrichment analysis (*Figure 2C*). Interestingly, we found that EGFR and PVR RTK signaling pathway is highly enriched in the oenocytes (*Figure 2C*; *Supplementary file 3*). While the expression of *Egfr* is specifically enriched in the adipose tissue (*Figure 2—figure supplement 1I-j*), we identify the Pvf1 receptor Pvr to be highly enriched in the oenocytes, albeit a mild enrichment seen in the other clusters (*Figure 2D*). We examined the distribution of PvR in the major metabolic tissues of interest:

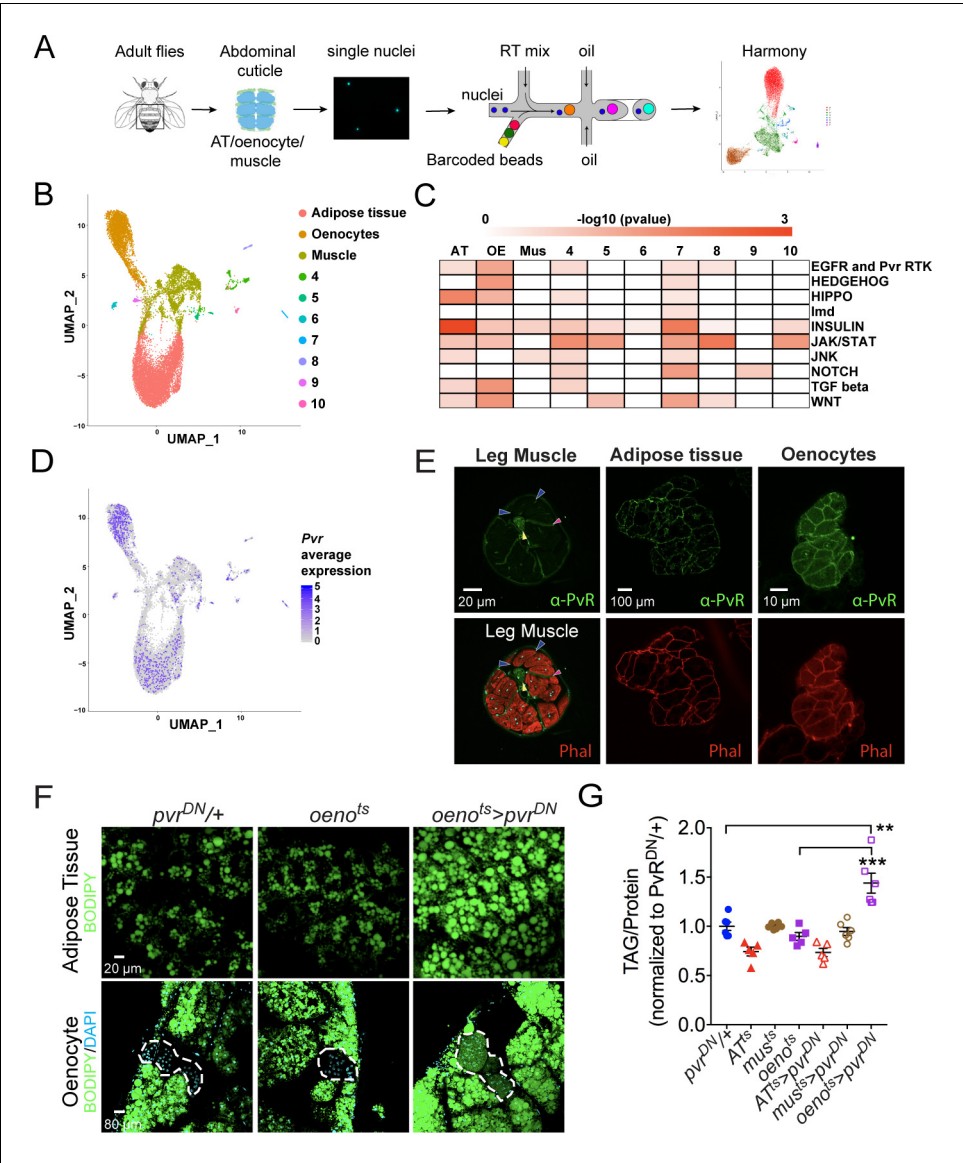

**Figure 2.** Single nuclei-RNA-sequencing reveals that oenocyte-specific PvR signaling protects against obesity. (**A**) Schematic of snRNA-seq workflow. Adult fly abdomens are dissected and dissociated to obtain high-quality single nuclei for downstream encapsulation by 10X genomics-based snRNA-seq platform and subsequent sequencing and analysis using Harmony. (**B**) Uniform Manifold Approximation and Projection (UMAP) plot representing 10 unique clusters identified while using a resolution of 0.1 for the clustering analysis. Each color and dot in the plot represent a cluster and a single nucleus, respectively. (**C**) Pathway enrichment analysis reveals EGFR and Pvr Receptor Tyrosine Kinase (RTK) signaling pathway enriched in oenocytes (OE) when compared to other clusters including adipose tissue (AT) and muscle (Mus). (**D**) UMAP plot representing the average expression of *Pvr*, which is highly enriched in oenocytes and to a lesser extent in adipose tissue and muscle. (**E**) Anti-PvR (Green) and phalloidin-594 (Red) staining of $w^{1118}$ (VDRC isogenic stock) adult male leg musculature (cross-section), adipose tissue, and oenocytes. Yellow and red arrowheads show the leg axon bundle and trachea respectively. Blue arrowheads show the sarcolemma of individual leg muscle bundles. (**F**) BODIPY staining showing neutral lipid accumulation in adult male adipose tissue and oenocytes (dorsal abdominal cuticle) of control males and males over-expressing $pvr^{DN}$ in the adult oenocytes ($oeno^{ts}>pvr^{DN}$). (**G**) Total TAG content of adult males over-expressing $pvr^{DN}$ in the adult adipose tissue ($AT^{ts}>pvr^{DN}$), muscle ($mus^{ts}>pvr^{DN}$), and oenocytes ($oeno^{ts}>pvr^{DN}$). Flies containing the UAS construct and the tissue-specific drivers alone serve as controls. N = 5–6, Cohorts of six males were used for each data point. One-way ANOVA followed by Tukey's HSD test, ** denotes p≤0.01, *** denotes p≤0.001, error bars = SEM.

The online version of this article includes the following figure supplement(s) for figure 2:

*Figure 2 continued on next page*

*Figure 2 continued*

**Figure supplement 1.** snRNA-seq of adult fly abdomens: validation of marker genes and top marker genes per cluster identified using a resolution of 0.1.
**Figure supplement 2.** PvR signaling works specifically in the oenocytes to protect against obesity.
**Figure supplement 3.** snRNA-seq analysis of the abdominal cuticle at higher resolution of 0.4.

muscle, adipose tissue, and the oenocytes, by immunostaining with an anti-PvR antibody (*Rosin et al., 2004*). Consistent with the prediction from the snRNA-seq analysis, PvR is present most prominently on the surface of the oenocytes followed by the adipose tissue cells (*Figure 2E*). In the leg muscles and indirect flight muscles, PvR localizes to the muscle sarcolemma (*Figure 2E* and *Figure 2—figure supplement 2B*), although the level of the protein on the sarcolemma of the leg muscles is relatively weak (*Figure 2E*).

## Oenocyte-specific loss of Pvf-Receptor (PvR) signaling leads to obesity

Since our data shows PvR and PvR-signaling to be enriched in the oenocytes, we asked whether PvR signaling in the oenocyte is necessary for protecting the adult flies against obesity. To test this possibility, we inhibited PvR signaling specifically in the oenocyte and determined the effect on whole animal TAG levels and lipid accumulation in the adipose tissue and oenocytes. Impairing PvR signaling in the adult oenocytes, by over-expressing a dominant negative form of the receptor, $pvr^{DN}$, ($oeno^{ts}$>$pvr^{DN}$) (*Brückner et al., 2004*) led to obesity phenotypes similar to $mus^{ts}$>$Pvf1$-i flies (*Figure 2F, G*). Similarly, impairing PvR signaling in the oenocyte by expressing an RNAi against $pvr$ ($oeno^{ts}$>$pvr$-i) also leads to obesity (*Figure 2—figure supplement 2D*). The obesity phenotype was also observed in $oeno^{ts}$>$pvr^{DN}$ female flies, indicating that the phenotype is not caused by loss of PvR signaling in the male accessory glands where the *PromE-Gal4* driver is also expressed (*Figure 2—figure supplement 2C*; *Billeter et al., 2009*). Surprisingly, over-expressing $pvr^{DN}$ in the adult adipose tissue and muscle did not lead to an obesity phenotype indicating that Pvf/PvR signaling is primarily required in the oenocytes to regulate lipid abundance (*Figure 2G* and *Figure 2—figure supplement 2A*). These results suggest that muscle-derived Pvf1 signals specifically to the oenocytes to regulate lipid content of the adipose tissue and steatosis in the oenocytes.

## Oenocyte-specific loss of TOR signaling leads to obesity

Downstream of PvR, Pvf signaling primarily activates the Ras/Raf/MEK/ERK pathway. To determine whether oenocyte-specific ERK signaling regulates lipid homeostasis, we measured neutral lipid storage in $oeno^{ts}$>$ERK$-i flies. Two independent and validated RNAi transgenes against ERK failed to replicate the obesity phenotype observed in $oeno^{ts}$>$pvr^{DN}$ flies, suggesting that PvR signaling in the oenocytes regulates lipid levels via an ERK-independent mechanism (*Figure 3A, C*). Previous studies in *Drosophila* S2 and Kc cells have shown that PvR can also activate the TOR pathway (*Sopko et al., 2015*; *Tran et al., 2013*). To test whether oenocyte-specific TOR signaling is involved in regulating lipid homeostasis, we inhibited TOR signaling in the oenocytes by over-expressing both $tsc1$ and $tsc2$ ($oeno^{ts}$>$tsc1,tsc2$). Similar to $mus^{ts}$>$pvf1$-i and $oeno^{ts}$>$pvr^{DN}$ flies, $oeno^{ts}$>$tsc1,tsc2$ flies showed massive accumulation of neutral lipids in both the adipose tissue and the oenocytes (*Figure 3B, C*).

Since PvR is a potent receptor tyrosine kinase, we further investigated whether PvR signaling can activate TOR signaling in the oenocyte via activation of Pi3K/Akt1 and regulate lipid homeostasis. The *Drosophila* genome encodes three Pi3Ks (Pi3K92E, Pi3K59F, and Pi3K68D) and one regulatory subunit (Pi3K21B). We knocked down each of the Pi3K components in the oenocytes and determined the effect on lipid accumulation in the oenocytes and adipose tissue (*Figure 3—figure supplement 1*). Oenocyte-specific loss of either Pi3K92E and the regulatory subunit Pi3K21B led to steatosis phenotypes (*Figure 3D,E*). Additionally, oenocyte-specific loss of *akt1* also led to steatosis phenotypes indicating that the Pi3K-Akt1 pathway in the oenocytes regulates lipid homeostasis (*Figure 3D, E*). Taken together, these results reveal that Pi3K/Akt1/TOR signaling in the oenocyte protects against obesity.

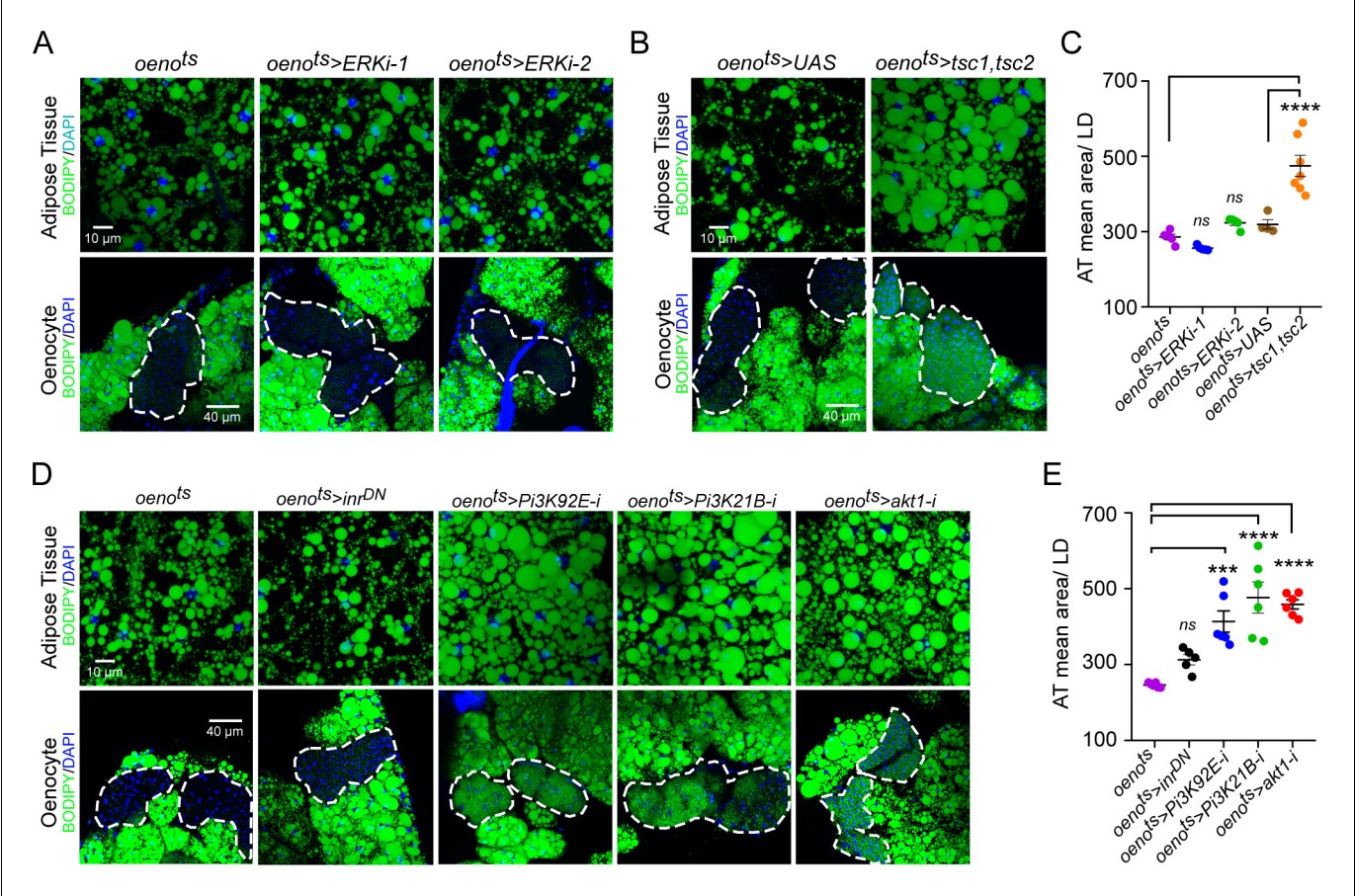

**Figure 3.** Oenocyte-specific Pi3K/Akt1/TOR signaling protects against obesity. (A) BODIPY staining showing neutral lipid accumulation in the adipose tissue and oenocytes of control males (*oeno^ts* = *oenocyte-Gal4^Gal80ts*) and males with oenocyte-specific knockdown of ERK (*oeno^ts>ERKi-1* and *oeno^ts>ERKi-2*) using two independent RNAi lines. (B) BODIPY staining showing neutral lipid accumulation in the adipose tissue and oenocytes of control males (*oeno^ts*) and males with oenocyte-specific over-expression of *tsc1* and *tsc2* (*oeno^ts>tsc1,tsc2*). (C) Mean lipid droplet size (≥10 microns in diameter) in the adipose tissue of flies shown in *Figure 2A and B*. Only oenocyte-specific loss of TOR signaling (*oeno^ts>tsc1,tsc2*) show a significant increase (p<0.001) compared to the control genotype. N = 6 animals, One-way ANOVA followed by Tukey's HSD test, **** denotes p≤0.0001, error bars = SEM. (D) BODIPY staining showing neutral lipid accumulation in the adipose tissue and oenocytes of control males (*oeno^ts*) and males with oenocyte-specific over-expression of *inr^DN* (*oeno^ts>inr^DN*) and oenocyte-specific (*oeno^ts>*) knockdown of *Pi3K92E* (*Dp110*), *Pi3K21B* (*Dp60*) and *akt1*. (E) Mean lipid droplet size (≥10 microns in diameter) in the adipose tissue of flies shown in *Figure 2D*. Oenocyte-specific (*oeno^ts>*) knockdown of *Pi3K92E* (*Dp110*), *Pi3K21B* (*Dp60*), and *akt1* lead to a significant increase (p<0.001 for *Pi3K92E,* and, p<0.0001 for *Pi3K21B* and *akt1*) compared to the control genotype. N = 6 animals, One-way ANOVA followed by Tukey's HSD test, *** denotes p≤0.001, **** denotes p≤0.0001, error bars = SEM.
The online version of this article includes the following figure supplement(s) for figure 3:

**Figure supplement 1.** Oenocyte-specific loss of only Pi3K92E and Pi3K21B leads to obesity.

## TOR signaling acts downstream of PvR in the oenocytes to regulate systemic lipid stores

We next analyzed whether PvR signaling in the oenocytes regulates lipid metabolism by activating the TOR pathway. For this, we first measured the levels of phospho-4EBP (p4EBP), a direct target of TOR, in the oenocytes of *mus^ts>pvf1-i* and *oeno^ts>pvr^DN* flies. Both *mus^ts>pvf1-i* flies and *oeno^ts>pvr^DN* flies showed a strong and significant down-regulation of p4EBP signal in the oenocytes compared to Gal4 alone controls (*Figure 4A, B, C*), indicating that muscle Pvf regulates TORC1 signaling in the oenocyte. Consistently, the extent of p4EBP down-regulation is similar to what is observed in *oeno^ts>tsc1,tsc2* flies that were used as a positive control for the assay (*Figure 4B, C*). Interestingly, oenocyte-specific loss of insulin receptor signaling (*oeno^ts>inr^DN*) failed to affect p4EBP levels (*Figure 4B , C*). In addition, we tested whether the obesity phenotype of *oeno^ts>pvr^DN* flies could be rescued by activating TOR signaling. To do this, we co-expressed *pvr^DN*

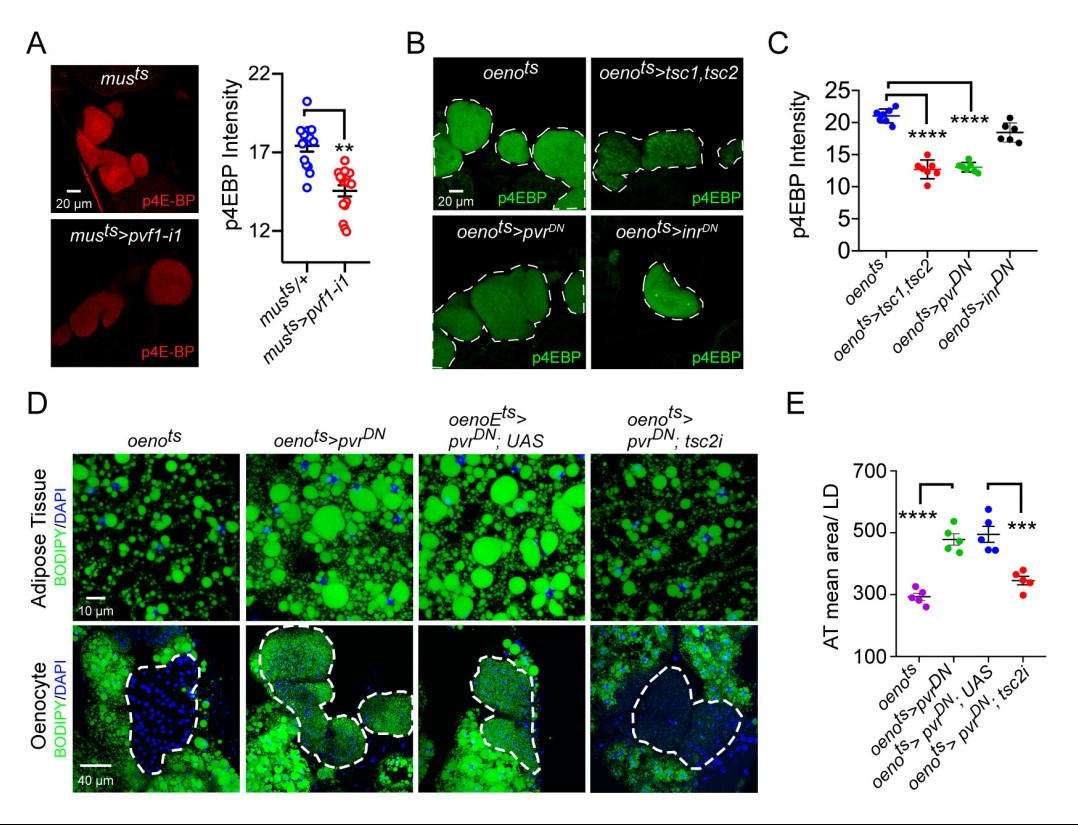

**Figure 4.** Pi3K/Akt1/TOR signaling acts downstream of PvR in the oenocytes to regulate adiposity. (**A**) p4EBP staining (Red) of the oenocytes from control ($mus^{ts}$ = muscle-$Gal4^{Gal80ts}$) and flies with muscle-specific knockdown of $pvf1$ ($mus^{ts}$>$pvf1$-$i1$). Muscle-specific loss of $pvf1$ leads to a significant decrease (p<0.01) in p4EBP levels in the oenocytes. N = 10 animals, student t-test, ** denotes p≤0.01, error bars = SEM. (**B**) p4EBP staining (Green) of oenocytes from control flies ($oeno^{ts}$ = oenocyte-$Gal4^{Gal80ts}$) and flies with oenocyte-specific over-expressing $tsc1$/$tsc2$ ($oeno^{ts}$>$tsc1,tsc2$), $pvr^{DN}$ ($oeno^{ts}$>$pvr^{DN}$) or $inr^{DN}$ ($oeno^{ts}$>$inr^{DN}$). (**C**) Quantification of p4EBP staining intensity in the oenocytes for samples shown in *Figure 3C*. Oenocyte-specific over-expression of $tsc1$/$tsc2$ ($oeno^{ts}$>$tsc1$, $tsc2$) and $pvr^{DN}$ ($oeno^{ts}$>$pvr^{DN}$) led to a significant reduction in p4EBP levels (p<0.0001). Over-expression of $inr^{DN}$ ($oeno^{ts}$>$inr^{DN}$), however, does not affect p4EBP levels significantly. N = 6/7 animals, Student t-test, **** denotes p≤0.0001, error bars = SEM. (**D**) BODIPY staining showing neutral lipid accumulation in the adipose tissue and oenocytes of control males ($oeno^{ts}$) and males with oenocyte-specific over-expression of either $pvr^{DN}$ ($oeno^{ts}$>$pvr^{DN}$) or over-expression of $pvr^{DN}$ along with $tsc2$ knockdown ($oeno^{ts}$>$pvr^{DN}$,$tsc2$-$i$). Flies over-expressing $pvr^{DN}$ in the oenocytes along with an empty UAS construct ($oeno^{ts}$>$pvr^{DN}$,$UAS$) serve as an additional control to account for any effect of Gal4 dilution on the obesity phenotype. (**E**) Mean lipid droplet size (≥10 microns in diameter) in the adipose tissue of flies shown in (**D**). Oenocyte-specific knockdown of $tsc2$ along with $pvr^{DN}$ over-expression ($oeno^{ts}$>$pvr^{DN}$,$tsc2$-$i$) significantly rescues the obesity phenotype observed flies with oenocyte-specific over-expression of $pvr^{DN}$ (p<0.001). N = 5 animals, One-way ANOVA followed by Tukey's HSD test, *** denotes p≤0.001, **** denotes p≤0.0001, error bars = SEM.

The online version of this article includes the following figure supplement(s) for figure 4:

**Figure supplement 1.** Oenocyte-specific activation of TOR signaling does not affect lipid accumulation.

**Figure supplement 2.** InR signaling or oenocyte size does not affect PvR signaling-mediated effects on systemic lipid accumulation.

**Figure supplement 3.** Additional control showing baseline lipid droplet size in the adipose tissue and lipid accumulation in the oenocytes.

---

and a $tsc2$-$RNAi$ transgene in the oenocytes ($oeno^{ts}$>$pvr^{DN}$; $tsc2$-$i$) and compared the lipid content of these flies to control fly lines. $tsc2$ knockdown strongly suppressed the obesity phenotype induced by oenocyte-specific expression of $pvr^{DN}$ (*Figure 4D, E*). Tsc1/2 regulates TOR signaling by suppressing the activity of Rheb which directly activates TORC1. We further investigated whether TORC1 functions downstream of PvR signaling to regulate lipid accumulation by testing whether co-

expression of a constitutively active form of *rheb* (*rheb^AV4*) can rescue the obesity phenotype observed in *oeno^ts^>pvr^DN* flies. Oenocyte-specific co-expression of *rheb^AV4* completely rescued the obesity phenotype observed in *oeno^ts^>pvr^DN* flies (*Figure 4—figure supplement 1A, B*). Altogether, these observations suggest that TOR signaling functions downstream of PvR in the oenocyte to negatively regulate lipid deposits.

Similar to vertebrates, the *Drosophila* TOR pathway can be activated by insulin/Pi3K/Akt signaling. To determine the potential involvement of oenocyte-specific insulin signaling in regulating lipid homeostasis, we over-expressed a dominant negative form of InR (*oeno^ts^>inr^DN*) and examined the effect on lipid accumulation. *oeno^ts^>inr^DN* flies did not show any increase in accumulation of lipids either in the adipose tissue or the oenocytes compared to control flies (*Figure 3D, E*). To further test the potential interaction between PvR signaling and InR signaling in regulating lipid accumulation, we co-expressed *inr^DN* and *pvr^DN* in oenocytes and determined the effect on lipid accumulation. Strikingly, while *oeno^ts^>pvr^DN; inr^DN* flies were still obese and showed excessive accumulation of lipids in the adipose tissue, lipid accumulation in oenocytes was suppressed (*Figure 4—figure supplement 2A*, Right most panels, and *Figure 4—figure supplement 2B*), indicating that InR signaling is required for lipid accumulation in the oenocytes. Interestingly, a previous report showed that activation of InR signaling in oenocytes led to accumulation of lipids both under fed and starved conditions (*Chatterjee et al., 2014*). However, whether InR signaling functions through TOR to activate lipid accumulation in the oenocyte was not tested. To further characterize the oenocyte-specific role of TOR on lipid accumulation, we activated TOR in the oenocytes (*oeno^ts^>tsc2-i*) and did not detect lipid accumulation either in the oenocytes or the adipose tissue (*Figure 4—figure supplement 1C, D*). Altogether, we demonstrate that in *Drosophila,* muscle-derived Pvf1 signals through PvR in the oenocyte to activate TOR, which in turn protects the animal against obesity.

## Muscle-to-Oenocyte Pvf1 signaling regulates lipid synthesis

*Drosophila* oenocytes are known to facilitate starvation-induced lipid mobilization in the *Drosophila* larvae and loss of this tissue leads to increased starvation sensitivity (*Gutierrez et al., 2007*). Similarly, in adult flies the oenocytes play a role in imparting starvation resistance by regulating production of very long chain fatty acids (VLCFAs) for waterproofing of the cuticle, especially when flies are starved under lower humidity conditions, and possibly by regulating lipid mobilization (*Chatterjee et al., 2014*; *Storelli et al., 2019*). We first investigated whether muscle-to-oenocyte Pvf1 signaling plays a role in starvation resistance. Compared to control flies, *mus^ts^>pvf1-i* and *oeno^ts^>pvr^DN* animals showed increased starvation resistance, suggesting that they are capable of mobilizing stored nutrients in response to starvation (*Figure 5A*). The improved starvation resistance of *mus^ts^>pvf1-i* and *oeno^ts^>pvr^DN* flies most likely reflects the fact that these animals had higher levels of stored TAGs and hence were able to use these reserves for a longer duration. As starvation can induce strong catabolic signals that can easily mask minor defects in lipid mobilization in *mus^ts^>pvf1-i* and *oeno^ts^>pvr^DN* flies, we measured the rate of lipid mobilization under steady state feeding conditions using radioisotope chasing. We labeled the TAG stores of control and experimental flies with [1-$^{14}$C]-Oleate for 3 days. Subsequently, we shifted the labeled flies to cold food and collected samples at 24, 48 and 72 hr post-transfer and measured $^{14}$C label in the TAG fractions using thin-layer chromatography (TLC). Interestingly, *mus^ts^>pvf1-i* flies showed similar rates of lipid mobilization from TAG stores compared to control flies (*Figure 5B*). Similarly, *oeno^ts^>tsc1, tsc2* and *oeno^ts^>pvr^DN* flies also showed comparable rates of lipid mobilization compared to control animals, indicating that loss of the muscle-to-oenocyte Pvf1 signaling axis does not impair lipid mobilization (*Figure 5C*). Since loss of the muscle-to-oenocyte Pvf1 signaling axis did not affect lipid mobilization, we tested whether flies lacking this pathway show increased lipid synthesis. To test this possibility, we transferred experimental and control animals to $^{14}$C-U-Sucrose containing food and measured the levels of $^{14}$C-incorporation over time in the TAG fraction of the flies using TLC. We found that *mus^ts^>pvf1-i*, *oeno^ts^>tsc1, tsc2*, and *oeno^ts^>pvr^DN* flies all showed an increased rate of $^{14}$C incorporation into TAG fractions compared to control animals (*Figure 5D, E*). In absence of any effects on lipid mobilization, the increased rate of lipid incorporation indicates an increased rate of lipid synthesis in the experimental flies.

Since loss of TOR signaling in the oenocytes led to increased lipid synthesis, we investigated the role of TOR signaling in regulating lipid synthetic genes in the oenocytes. We extracted total RNA from adult oenocyte/adipose tissue complexes and measured the expression levels of two oenocyte-

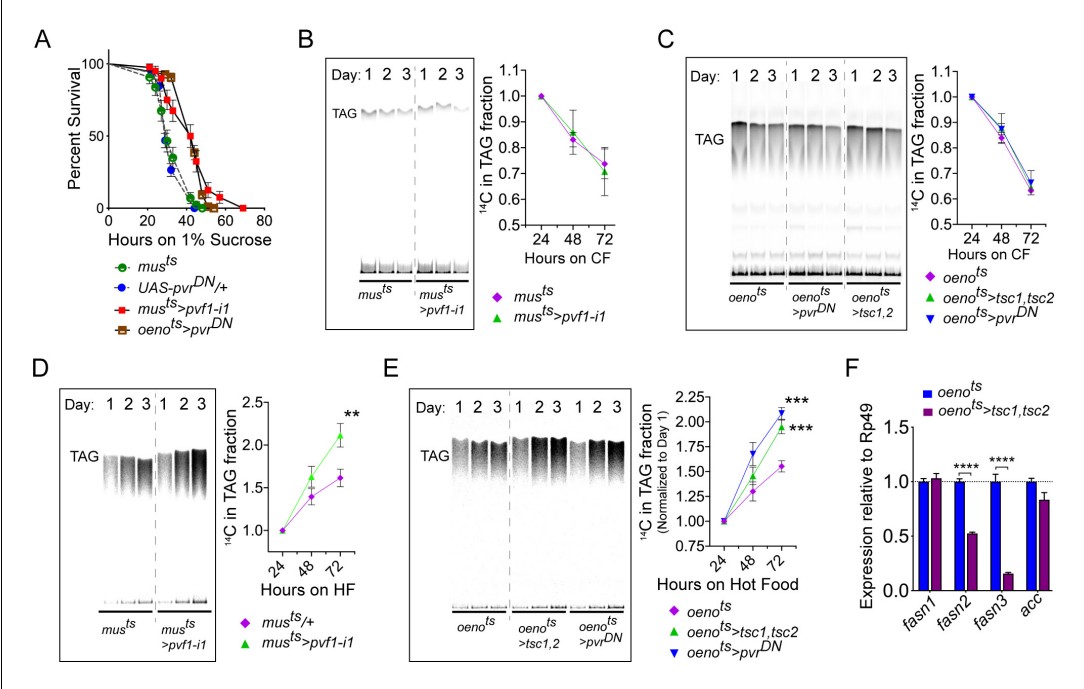

**Figure 5.** Muscle to oenocyte Pvf1 signaling suppresses lipid synthesis. (A) Starvation resistance of adult male flies on 1% sucrose and 0.8% agar food. Males with muscle-specific knockdown of *pvf1* (*mus^ts^>pvf1-i1*) and oenocyte-specific loss of PvR signaling (*oeno^ts^>pvr^DN^*) were significantly more resistant to starvation compared to respective controls flies (*mus^ts^ = muscle-Gal4^Gal80ts^* and *UAS-pvr^DN^/+*). N = 100 (animals), Log rank (Mantel-Cox) test, p≤0.0001 for both comparisons, error bars = SEM. (B) Rate of lipid mobilization in control (*mus^ts^*) and flies with muscle-specific loss of *pvf1* (*mus^ts^>pvf1-i1*). N = 4, Cohorts of 23 adult males were used per data point. Multiple Student t-test, error bars = SEM. Note: The control and experimental TLC bands were rearranged for representation from the same TLC plate. Junctions are marked with a dotted line. (C) Rate of lipid mobilization in control (*oeno^ts^ = oenocyte-Gal4^Gal80ts^*) and flies with oenocyte-specific loss of TOR signaling (*oeno^ts^>tsc1,tsc2*) and PvR signaling (*oeno^ts^>pvr^DN^*). N = 4, Cohorts of 23 adult males were used per data point. Multiple Student t-test, error bars = SEM. Note: The control and experimental TLC bands were rearranged for representation from the same TLC plate. Junctions are marked with a dotted line. (D) Rate of lipid synthesis and incorporation from [U-^14^C]-Sucrose in control (*mus^ts^*) and flies with muscle-specific loss of *pvf1* (*mus^ts^>pvf1-i1*). Flies lacking Pvf1 in the muscle show a significantly faster rate of lipid incorporation compared to control animals. N = 4, Cohorts of 23 adult males were used per data point. ** denotes p≤0.01 at 72 hr on hot food. Multiple student t-test, error bars = SEM. Note: The control and experimental TLC bands were rearranged for representation from the same TLC plate. Junctions are marked with a dotted line. (E) Rate of lipid synthesis and incorporation from [U-^14^C]-Sucrose in control flies (*oeno^ts^*) and flies with oenocyte-specific loss of TOR signaling (*oeno^ts^>tsc1,tsc2*) and PvR signaling (*oeno^ts^>pvr^DN^*). Flies lacking TOR or PvR signaling in the oenocytes show a significantly faster rate of lipid incorporation compared to control animals. N = 4, Cohorts of 23 adult males were used per data point. *** denotes p≤0.001 at 72 hr on hot food. Multiple student t-test, error bars = SEM. Note: The control and experimental TLC bands were rearranged for representation from the same TLC plate. Junctions are marked with a dotted line. (F) Expression level of key lipid synthesis genes in control (*oeno^ts^*) and flies with oenocyte-specific loss of TOR signaling (*oeno^ts^>tsc1,tsc2*). Only oenocyte-specific fatty acid synthases, *fasn2* and *fasn3*, show a significant reduction in expression in the experimental flies. N = 4, Cohorts of 23 adult males were used per data point. One-way ANOVAa followed by Tukey's HSD test, **** denotes p≤0.0001, error bars = SEM.

specific fatty acid synthases, *fasn2* and *fasn3*. In addition, we measured the expression of two lipogenic genes, *acc* and *fasn1*, that are not exclusively expressed in the adult oenocytes in *Drosophila*. While expression levels of *acc* and *fasn1* did not change in *oeno^ts^>tsc1,tsc2* flies, expression levels of *fasn2* and *fasn3* were strongly downregulated (*Figure 5F*). This indicates that loss of TOR signaling downregulates lipogenic genes in the oenocytes, and that the increased lipid synthesis observed in *oeno^ts^>tsc1,tsc2* flies is caused by a mechanism independent of the role of TOR in regulating the expression of lipogenic genes in the oenocytes.

## Muscle-to-Oenocyte Pvf1 signaling regulates post-eclosion restoration of stored lipids in adult adipose tissue

When adult flies emerge from their pupal cases the adult adipose tissue has very low stored lipid content (*Storelli et al., 2019*), and adipose tissue cells of post-eclosion flies are also notably small in size (*Figure 6—figure supplement 1*, left most panels). Over the course of the next 3 to 7 days the

adipose tissue builds up its lipid stores through feeding and de novo lipid synthesis and expands significantly in size both at cellular and tissue levels (*Figure 6—figure supplement 1*, middle and right most panels). While the average size of the lipid droplets does not change drastically during this period, the number of lipid droplets per cell increases drastically and a large number of smaller lipid droplets start appearing in the adipose tissue cells (*Figure 6—figure supplement 1*). These observations suggest that build up of adipose tissue lipid stores happens by formation of new lipid droplets that become bigger in size as adipose tissue lipid build up progresses. Since muscle-to-oenocyte Pvf1 signaling axis negatively regulates lipid synthesis, we hypothesized that this pathway is needed to inhibit lipid synthesis once the build up of adipose tissue lipid stores reaches completion. Consistent with this hypothesis, we find that muscle-specific expression levels of *pvf1* is low in newly eclosed flies and increases rapidly over the course of the next 7 days (*Figure 6A*). To further test the hypothesis that muscle-Pvf1 limits the extent of TAG build up in the adipose tissue of newly eclosed flies, we over-expressed *pvf1* in the adult muscle from day 1 of eclosion and measured the frequency of large lipid droplets (LD $\geq$ 5 µm in diameter/cell) per cell in the adult adipose tissue using BODIPY staining. Compared to controls, *mus^{ts}>pvf1* flies tend to accumulate much lower number of large lipid droplets per adipose tissue cell (*Figure 6B, D*). Additionally, the experimental animals tend to have larger number of empty adipose tissue cells per animal compared to control flies (*Figure 6C*). These results suggest that muscle-derived Pvf1 helps terminate the adipose tissue lipid build up phase by suppressing lipid synthesis by signaling to the *Drosophila* oenocytes.

## Discussion

The presence in vertebrates of multiple PDGF/VEGF signaling ligands and cognate receptors makes it difficult to assess their roles in inter-organ communication. Additionally, understanding the tissue-specific roles of these molecules, while circumventing the critical role they play in regulating tissue vascularization, is equally challenging in vertebrate models. Here, we investigated the tissue-specific roles of the ancestral PDGF/VEGF-like factors and the single PDGF/VEGF-receptor in *Drosophila* in lipid homeostasis. Our results demonstrate that in adult flies the PDGF/VEGF like factor, Pvf1, is a muscle-derived signaling molecule (myokine) that suppresses systemic lipid synthesis by signaling to the *Drosophila* hepatocyte-like cells/oenocytes.

The *Drosophila* larval and adult adipose tissues have distinct developmental origins. The larval adipose tissue undergo drastic morphological changes during metamorphosis and dissociate into individual large spherical cells (*Nelliot et al., 2006*). These free-floating adipose cells persist to the young adult stage where they play a crucial role in protecting the animal from starvation and desiccation (*Aguila et al., 2007*; *Storelli et al., 2019*). These larval adipose tissue cells are ultimately removed via cell death (*Aguila et al., 2007*). Adult-specific adipose tissue cells develop during the pupal stage from yet unknown progenitor cells and have very little lipid stores in newly eclosed flies (*Figure 6—figure supplement 1*). Over the period of next 3–5 days the adult adipose tissue builds up its lipid reserves through feeding and de-novo lipid synthesis. However, at the end of the lipid build-up phase, the rate of lipid synthesis must be suppressed to avoid over-loading of the adipose tissue and prevent lipid toxicity. Our data suggests that muscle Pvf1 signaling suppresses lipid synthesis at the end of the adult adipose tissue lipid build-up phase. Pvf1 production in the adult muscles peaks around the time when adult adipose tissue lipid stores reach steady state capacity. In turn, muscle-derived Pvf1 suppresses lipid synthesis and lipid incorporation by activating TOR signaling in the oenocytes.

### *Drosophila* Pvf1 functions as a myokine that suppresses lipid synthesis

Our study reveals that Pvf1 is abundant in the tubular muscles of the *Drosophila* leg and abdomen. In these striated muscles, the protein localizes between individual myofibrils and is particularly enriched at the M and Z bands. *Drosophila* musculature can be broadly categorized into two groups, the fibrillar muscles and the tubular muscles, with distinct morphological and physiological characteristics. *Drosophila* IFMs of the thorax belong to the fibrillar muscle group and are stretch-activated, oxidative, slow twitch muscles that are similar to vertebrate cardiac muscles (*Schönbauer et al., 2011*). By contrast, *Drosophila* leg muscles and abdominal muscles belong to the tubular muscle group. These muscles are striated, $Ca^{2+}$ activated, and glycolytic in nature. The tubular muscles are structurally and functionally closer to vertebrate skeletal muscles (*Schnorrer et al., 2010*;

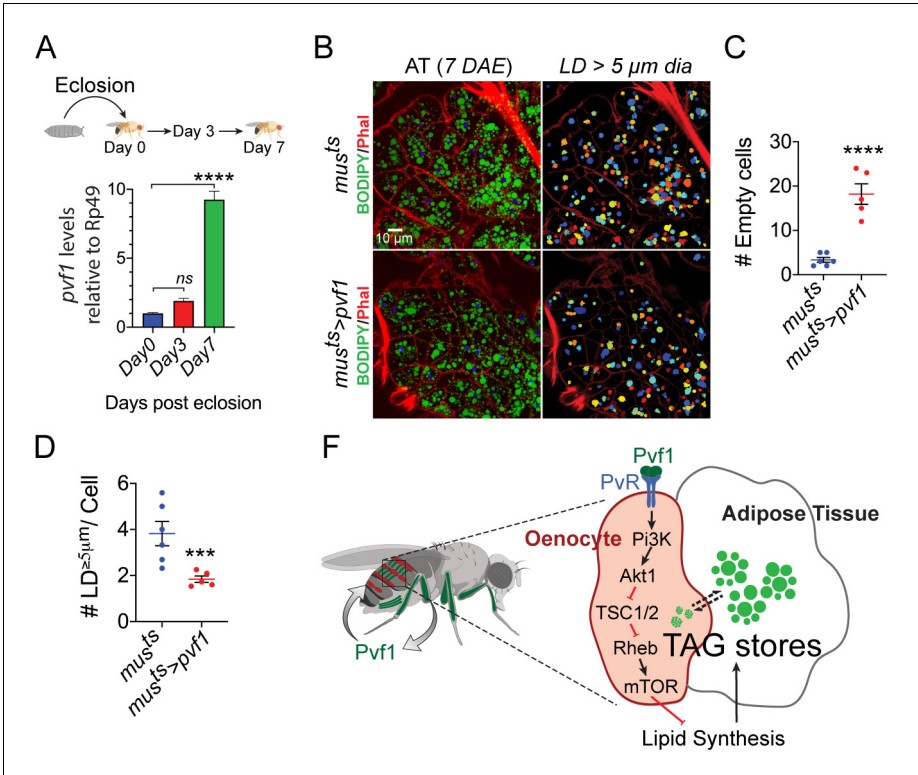

**Figure 6.** Muscle-Pvf1 limits post-eclosion lipid recovery by suppressing lipid synthesis. (**A**) Muscle-specific expression of *pvf1* in freshly eclosed *w*[1118] males on days 0, 3 and 7 post eclosion. N = 4, 10 adult male abdominal cuticles were used per data point. One-way Anova followed by Dunnett multiple comparison test, *ns* denotes Not Significant with p=*0.202*, **** denotes p≤*0.0001*, error bars = SEM. (**B**) BODIPY staining of adipose tissue from 7-day-old control males (*mus*[ts] = *muscle-Gal4*[Gal80ts]) and males with muscle-specific over-expression of *pvf1* (*mus*[ts]>*Pvf1*) from Day 0 of eclosion. Panels on the right show identification of lipid droplets (LD) that are ≥ 5 µm in diameter using Cell Profiler. DAE = days after eclosion. (**C**) Quantification of number of adipose tissue cells that do not contain LDs that are ≥ 5 µm in diameter for control (*mus*[ts]) flies and flies with muscle-specific over-expression of *pvf1* (*mus*[ts]>*pvf1*). N = 5/6 animals, Student t-test, **** denotes p≤*0.0001*, error bars = SEM. (**D**) Quantification of number of large (≥5 µm in diameter) LDs per cell in control (*mus*[ts]) flies and flies with muscle-specific over-expression of *pvf1* (*mus*[ts]>*pvf1*). N = 5/6 animals, Student t-test, *** denotes p≤*0.001*, error bars = SEM. (**E**) Model of the role of muscle-derived Pvf1 in regulating systemic lipid content by signaling to the oenocytes of adult male flies.

The online version of this article includes the following figure supplement(s) for figure 6:

**Figure supplement 1.** Recovery of adipose tissue lipid stores involve formation and expansion of new lipid droplets, and, expansion of the adipose tissue cells.

**Figure supplement 2.** Oenocyte-specific activation of lipid synthesis leads to increased lipid stores in the adipose tissue.

---

*Schönbauer et al., 2011*). Expression of Pvf1 in the tubular muscles of the *Drosophila* leg may reflect a potentially conserved role of this molecule as a skeletal-muscle-derived myokine. The fact that most of the myokines in vertebrates were identified in striated skeletal muscles supports this possibility (*Pedersen and Febbraio, 2012*; *So et al., 2014*). Moreover, vertebrate VEGF ligands, VEGF-A and VEGF-B, have also been shown to be stored and released from skeletal muscles (*Boström et al., 2012*; *Vind et al., 2011*).

Interestingly, in vertebrates, the expression and release of VEGF ligands are regulated by muscle activity (*Boström et al., 2012*; *Vind et al., 2011*). In mice, expression of VEGF-B in the skeletal muscles is regulated by PGC1-α, one of the key downstream effectors of muscle activity. Additionally, expression of VEGF-B is upregulated in both mouse and human skeletal muscles in response to muscle activity (*Boström et al., 2012*; *Vind et al., 2011*). Similarly, expression of VEGF-A is induced by muscle contraction (*Boström et al., 2012*). We did not see any effect of muscle activity on the

expression levels of *pvf1* in the *Drosophila* muscles. We also could not demonstrate whether muscle activity regulates release of Pvf1 primarily due to the difficulty in collecting adequate amounts of hemolymph from the adult males. However, the localization of Pvf1 to the M/Z bands suggests a potential role for muscle activity in Pvf1 release. The M and Z bands of skeletal muscles are important centers for sensing muscle stress and strain. These protein-dense regions of the muscle house a number of proteins that can act as mechano-sensors and mediate signaling events including translocation of selected transcription factors to the nucleus (*Hoshijima, 2006*; *Lange et al., 2020*; *Miller et al., 2003*). Pvf1, therefore, is ideally located to be able to sense muscle contraction and be released in response to muscle activity. Further work, contingent on the development of new tools and techniques, will be necessary to measure Pvf1 release into the hemolymph and study the regulation of this release by exercise.

We have previously shown that Pvf1 released from gut tumors generated by activation of the oncogene *yorkie* leads to wasting of *Drosophila* muscle and adipose tissue (*Song et al., 2019*). Adipose tissue wasting in these animals is characterized by increased lipolysis and release of free fatty acids (FFAs) in circulation. However, we did not observe any role of Pvf signaling in regulating lipolysis in the adipose tissue of healthy well-fed flies without tumors. Loss of PvR signaling in the adipose tissue did not have any effect on lipid content. Additionally, over-expressing Pvf1 in the muscle did not lead to the bloating phenotype commonly seen in cachectic animals with gut tumors (*Kwon et al., 2015*; *Song et al., 2019*). We conclude that Pvf1 affects wasting of the adipose tissue only in the context of gut tumors and that the effect could involve the following mechanisms: (1) the gut tumor releases pathologically high levels of Pvf1 into circulation leading to ectopic activation of PvR signaling in the adipose tissue, and, that such high levels of Pvf1 are not released by the muscle (even when *pvf1* is over-expressed in the muscle); (2) Pvf1 causes adipose tissue wasting in the context of other signals that emanate from the gut tumor that are not available in flies that do not have tumors.

## *Drosophila* oenocytes regulate lipid synthesis and lipid content of the adipose tissue

Only oenocyte-specific loss of PvR signaling phenocopies the obesity phenotype caused by muscle-specific loss of Pvf1, indicating that muscle-Pvf1 primarily signals to the oenocytes to regulate systemic lipid content. Additionally, muscle-specific loss of Pvf1, as well as oenocyte-specific loss of PvR and its downstream effector TOR, leads to an increase in the rate of lipid synthesis. These observations indicate a role for the *Drosophila* oenocytes in lipid synthesis and lipid accumulation in the adipose tissue. Oenocytes have been implicated in lipid metabolism previously and these cells are known to express a diverse set of lipid metabolizing genes including but not limited to fatty acid synthases, fatty acid desaturases, fatty acid elongases, fatty acid β-oxidation enzymes and lipophorin receptors (reviewed in *Makki et al., 2014*). Functionally, the oenocytes are proposed to mediate a number of lipid metabolism processes. Oenocytes tend to accumulate lipids during starvation (presumably for the purpose of processing lipids for transport to other organs and generation of ketone bodies) and are necessary for starvation induced mobilization of lipids from the adipose tissue (*Gutierrez et al., 2007*; *Makki et al., 2014*). This role is similar to the function of the liver in clearing FFAs from circulation during starvation for the purpose of ketone body generation, and redistribution of FFAs to other organs by converting them to TAG and packaging into very-low density lipoproteins. However, our [1-$^{14}$C]-oleate chase assay did not show any effect of oenocyte-specific loss of PvR/TOR signaling on the rate of lipid utilization, indicating that this pathway does not affect oenocyte-dependent lipid mobilization.

Oenocytes also play a crucial role in the production of VLCFAs needed for waterproofing of the cuticle (*Storelli et al., 2019*). Results of our starvation resistance assay indicate that loss of the muscle-to-oenocyte Pvf1 signaling axis does not affect waterproofing of the adult cuticle. Storelli et al. have recently shown that the lethality observed in traditionally used starvation assays is largely caused by desiccation unless the assay is performed under saturated humidity conditions. Since our starvation assay was performed under 60% relative humidity (i.e. non-saturated levels), it is likely that desiccation played a partial role in causing starvation-induced lethality. Any defects in waterproofing of the adult cuticle would have led to reduced starvation resistance (*Nguyen et al., 2008*). However, both muscle-specific loss of Pvf1 and oenocyte-specific loss of PvR led to increased starvation resistance suggesting normal waterproofing in these animals. The increased starvation resistance in these

animals is likely the result of these animals having higher stored lipid content that helps them to survive longer without food.

Insect oenocytes were originally believed to be lipid synthesizing cells because they contain wax-like granules (*Makki et al., 2014*). These cells express a large number of lipid-synthesizing genes and the abundance of smooth endoplasmic reticulum further suggest a role for this organ in lipid synthesis and transport (*Chatterjee et al., 2014*; *Jackson and Locke, 1989*; *Wigglesworth, 1988*). However, evidence for potential involvement of the oenocytes in regulating lipid synthesis and lipid deposition in the adipose tissue has been lacking. The fact that two of the three fatty acid synthases (*fasn2* and *fasn3*) encoded by the *Drosophila* genome are expressed specifically in adult oenocytes suggests a potential role for these cells in lipid synthesis (*Chung et al., 2014*). Our observation that oenocyte-specific loss of PvR and its downstream effector TOR leads to increased lipid synthesis and increased lipid content of the adipose tissue strongly supports this possibility. Our data further suggests that involvement of the oenocytes in mediating lipid synthesis is more pronounced in newly eclosed adults when the adipose tissue needs to actively build up its lipid stores. In later stages of life, the lipid synthetic role of the oenocytes is repressed by the muscle-to-oenocyte Pvf1 signaling axis. Our observation also raises the question of whether FFAs made in the oenocytes can be transported to the adipose tissue for storage. We tested this possibility by over-expressing the lipogenic genes *fasn1* and *fasn3*, which regulate the rate limiting steps of de-novo lipid synthesis, in the oenocytes. We found that excess lipids made in the oenocytes do end up in the adipose tissue of the animal leading to increased lipid droplet size in the adipose tissue (*Figure 6—figure supplement 2*). Taken together, these results provide evidence for the role of *Drosophila* oenocytes in lipid synthesis and storage of neutral lipids in the adipose tissue of the animal. Interestingly, the vertebrate liver is also one of the primary sites for de-novo lipid synthesis and lipids synthesized in the liver can be transported to the adipose tissue for the purpose of storage (*Gibbons et al., 2000*; *Meex and Watt, 2017*). Hence, the fundamental role of the oenocytes and the mammalian liver converge with respect to their involvement in lipid synthesis.

## A unique role of oenocyte-specific TOR signaling in lipid synthesis

We observed that oenocyte-specific loss of the components of the Pi3K/Akt1/TOR signaling pathway leads to increased lipid synthesis. The increased rate of lipid synthesis in flies lacking TOR signaling in the oenocytes is paradoxical to our current knowledge of how TOR signaling affects expression of lipid synthesis genes. In both vertebrates and flies, TOR signaling is known to facilitate lipid synthesis by inducing the expression of key lipid synthesis genes such as *acetyl CoA-carboxylase* and *fatty acid synthase* via activation of SREBP-1 proteins (*Han and Wang, 2018*; *Heier and Kühnlein, 2018*; *Porstmann et al., 2008*). We therefore checked how oenocyte-specific loss of TOR signaling affects expression of oenocyte-specific fatty acid synthases (*fasn2* and *fasn3*) and oenocyte non-specific fatty acid synthesis genes (*fasn1* and *acc*). We observed that oenocyte-specific loss of TOR strongly down-regulates only *fasn2* and *fasn3*, while the expression of adipose tissue specific *fasn1* and *acc* did not change, indicating that TOR signaling is required for the expression of lipogenic genes in the oenocytes. An increase in lipid synthesis in response to loss of TOR in the oenocytes is quite intriguing and the mechanism remains to be addressed. We do not think the increase in lipid synthesis happens in the oenocytes since loss of TOR signaling rather reduces expression of lipogenic genes in the oenocytes. The increase in lipid synthesis could happen either as a result of compensatory upregulation of lipid synthesis in the adipose tissue or due to disruption of an as yet unknown role of the oenocytes in lipid synthesis that hinges on TOR signaling. The fact that the expression levels of *fasn1* and *acc* does not change significantly in animals lacking TOR signaling in the oenocytes indicates that compensatory upregulation of lipid synthesis, if present, does not happen through transcriptional upregulation of lipid synthesis genes in the adipose tissue. It is still possible, however, that the increase in lipid synthesis is caused by post-translational modifications of the enzymes. Alternatively, loss of TOR in the oenocyte could affect tissue distribution of lipids or impair clearing of dietary lipids via formation of cuticular hydrocarbons. Understanding the tissue specific alterations in gene expression and changes in the phosphorylation states of key lipogenic proteins in the adipose tissue of animals lacking TOR signaling in oenocytes could shed more light on the mechanisms involved.

Interestingly, our data suggests that the *Drosophila* InR does not play a role in activating TOR signaling in the oenocytes. While loss of TOR signaling in the oenocytes leads to obesity, loss of InR

signaling does not. Additionally, loss of oenocyte specific InR signaling did not have any effect on p4EBP levels in oenocytes. Moreover, InR signaling and TOR signaling also diverge in their roles in regulating the size of oenocytes. While loss of InR signaling leads to a significant reduction in the size of oenocytes, loss of TOR does not. Further suggesting that TOR does not act downstream of InR in oenocytes. Rather, our data suggests that in wildtype well-fed flies TOR signaling in oenocytes is activated by the Pvf receptor. Interestingly, insulin dependent activation of TOR is not universal. For instance, in the specialized cells of non-obese mouse liver, InR does not play any role in activation of TOR and downstream activation of SREBP-1c (*Haas et al., 2012*).

*Drosophila* larval oenocytes are known to accumulate lipids in response to starvation (*Gutierrez et al., 2007*). Chatterjee et al. also showed that starving adult females for 36 hr is capable of inducing lipid accumulation in the oenocytes and that this response is dependent of InR signaling (*Chatterjee et al., 2014*). Since TOR signaling is a known metabolic regulator, one alternate hypothesis that could explain some of our data is that loss of PvR/TOR signaling leads to a starvation like response specifically in the oenocyte leading to InR-dependent accumulation of lipid droplets. To address this possibility, we performed single nuclei sequencing of the adult male abdominal cuticle (and the tissues residing within) derived from *oeno^ts^>tsc1,tsc2* flies. The animals were raised under identical experimental conditions as control animals reported in *Figure 2*. We then re-analyzed the two snRNA-seq data sets after correcting for batch effects using harmony. The resulting UMAP plots for both genotypes look similar to our original UMAP plot for the control flies and identifies all the clusters reported in *Figure 2* (*Figure 7A, B*). The percentage of nuclei that constitute each of our major clusters remained similar in both genotypes and the top marker genes for each of the clusters did not change (*Figure 7C, D*). We subsequently converted the oenocyte-specific gene expression profiles from both data sets to pseudobulk expression for the genes that were detected. This allowed us to compare the expression profiles of the oenocytes from control animals and animals lacking TOR signaling in oenocytes (*Figure 7E*). We specifically looked at the effect of losing TOR on the expression of the 47 genes that Chatterjee et al. had reported to be up-regulated in oenocytes in response to starvation. Thirty-six of these genes were detected by our single nuclei sequencing analysis, however, none of them changed significantly (we used a cutoff of 1.5-fold change) (*Figure 7E*, *Supplementary file 4*). Based on this observation, we conclude that loss of TOR signaling most likely does not mount a starvation like response in the oenocytes.

Since Pi3K, Akt, and TOR signaling also control cellular growth, there is a possibility that manipulating these genes lead to changes in oenocyte size that in turn leads to the lipid accumulation phenotypes we observe. Indeed, we saw changes in oenocyte size when we disrupted some of these pathways. Since all our studies involved conditional knockdown or over-expression in post-developmental adult animals, the effect on size is most likely caused by changes in the size of individual oenocytes. Consistently, we find that genotypes that lead to smaller oenocytes tend to have more densely packed nuclei. We therefore used nuclei density as a readout to quantify the effect of losing PvR, TOR, Akt1, or InR signaling on oenocyte size. Loss of PvR signaling did not have any effect on oenocyte size (*Figure 4—figure supplement 2C, D*) and loss of TOR signaling leads to a small but not significant increase in oenocyte size. Whereas loss of *InR* signaling or *akt1* leads to significant decrease in oenocyte size. However, we did not observe any clear correlation between oenocyte size and the obesity phenotype (*Figure 4—figure supplement 2C, D*). Thus, while oenocyte size could potentially affect lipid metabolism, at least in our study, the size of the oenocytes was not the driving force for the phenotypes observed.

Serum levels of VEGF-A is high in obese individuals and drops rapidly in response to bariatric surgery, suggesting a role for VEGF-A in obesity (*García de la Torre et al., 2008*; *Loebig et al., 2010*; *Silha et al., 2005*). However, evidence on whether VEGF-A or other VEGFs are deleterious vs beneficial in the context of the pathophysiology of obesity is unclear. Adipose tissue-specific over-expression of both VEGF-B and VEGF-A has been shown to improve adipose tissue vascularization, reduce hypoxia, induce browning of fat, increase thermogenesis, and protect against obesity (*Elias et al., 2012*; *Robciuc et al., 2016*; *Sun et al., 2012*; *Sung et al., 2013*). At the same time, blocking VEGF-A signaling in the adipose tissue of genetically obese mice leads to reduction of body weight gain, improvement in insulin sensitivity, and a decrease in adipose tissue inflammation (*Sun et al., 2012*). Moreover, systemic inhibition of VEGF-A or VEGF-B signaling by injecting neutralizing monoclonal antibodies have also shown remarkable effects in improving insulin sensitivity in the muscle, adipose tissue, and the liver of high-fat diet-induced mouse models of obesity and diabetes

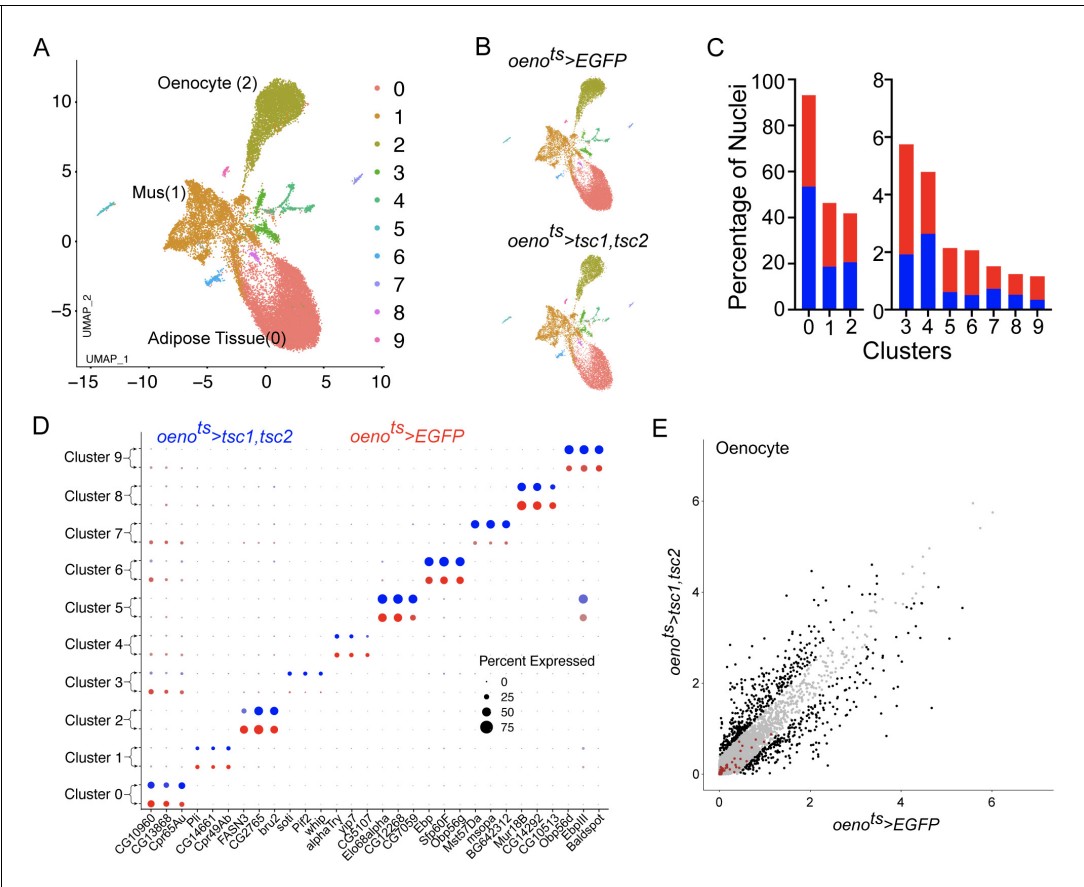

**Figure 7.** Differential snRNA-seq of the abdominal cuticle upon oenocyte-specific loss of TOR. (**A**) Integrated Uniform Manifold Approximation and Projection (UMAP) plot of snRNA-seq data derived from the abdominal cuticle samples pertaining to oenocyte-specific overexpression of *tuberous sclerosis complex 1* and *2* (*tsc1, tsc2*), which results in the blockade of TOR activity. The UMAP shows 10 distinct clusters (0–9) with the Adipose (0), Muscle (1), and Oenocytes (2) as the major tissue clusters. *oeno*$^{ts}$ = oenocyte-Gal4$^{Gal80ts}$. (**B**) UMAPs separated (from A) based on samples oenocyte-specific overexpression of EGFP control (*oeno*$^{ts}$>*EGFP*) and *tsc1, tsc2* (*oeno*$^{ts}$>*tsc1, tsc2*). (**C**) Bar graphs depicting the percentages of nuclei between the two genotypes *oeno*$^{ts}$ >*EGFP* (red) and *oeno*$^{ts}$>*tsc1, tsc2* (blue). Note that the fraction of nuclei did not change in cluster 2, which represents the Oenocytes. (**D**) Dot plot representing top three genes enriched per cluster based on average expression (logFC). Color gradient of the dot represents the expression level, while the size represents percentage of cells expressing any gene per cluster. The dots within each cluster are separated based on the two genotypes *oeno*$^{ts}$>*EGFP* (red) and *oeno*$^{ts}$>*tsc1, tsc2* (blue) to showcase changes in gene expression, if any. (**E**) Scatter plot depicting the differentially expressed genes within the Oenocyte cluster pertaining to the two genotypes, *oeno*$^{ts}$>*EGFP* and *oeno*$^{ts}$>*tsc1,tsc2* [log (average expression +1)]. The differential gene expression analysis reveals that certain genes known to be induced upon starvation (highlighted as red dots) remain unchanged upon loss of TOR activity within oenocytes compared to controls. See ***Supplementary file 4*** for the list of starvation-induced genes.

(***Hagberg et al., 2012***; ***Wu et al., 2014***). Although the evidence on the roles of VEGF/PDGF signaling ligands in obesity and insulin resistance is well established, the mechanisms clearly are quite complex and are often context dependent. Consequently, a wider look at various tissue specific roles of PDGF/VEGF signaling will be necessary to comprehensively understand the roles of PDGF/VEGF signaling in lipid metabolism. Our work demonstrates an evolutionarily conserved role for PDGF/VEGF signaling in lipid metabolism and a non-endothelial cell dependent role of the signaling pathway in lipid synthesis. Additionally, our findings suggest an atypical tissue-specific role of TOR signaling in suppressing lipid synthesis at the level of the whole organism. Further studies will be required to determine whether vertebrate VEGF/PDGF and TOR signaling exerts similar roles either in the vertebrate liver or in other specialized organ.

### A transcriptomic resource of adipose/oenocyte/muscle tissues

We made use of snRNA-Seq technology to identify expression of Pvr precisely in certain tissues in the complex abdominal region, which harbors several metabolically active tissues including adipose tissues, oenocytes, and muscle in *Drosophila*. As yet, there is no systematic study of a complete transcriptomics resource of each of these tissues considering the difficulty in dissecting and segregating these tissues for downstream sequencing. Thus, our study also provides a rich resource of gene expression profiles, paving way for a systems-level understanding of each of these tissues in *Drosophila*.

## Materials and methods

**Key resources table**

| Reagent type (species) or resource | Designation | Source or reference | Identifiers | Additional information |
|---|---|---|---|---|
| Gene (*D. melanogaster*) | *Pvf1* | FlyBase | FLYB: FBgn0030964 | NA |
| Gene (*D. melanogaster*) | *Pvf2* | FlyBase | FLYB: FBgn0031888 | NA |
| Gene (*D. melanogaster*) | *Pvf3* | FlyBase | FLYB: FBgn0085407 | NA |
| Gene (*D. melanogaster*) | *Pvr* | FlyBase | FLYB: FBgn0085407 | NA |
| Gene (*D. melanogaster*) | *Pi3K21B* | FlyBase | FLYB: FBgn0020622 | NA |
| Gene (*D. melanogaster*) | *Pi3K92E* | FlyBase | FLYB: FBgn0015279 | NA |
| Gene (*D. melanogaster*) | *Akt1* | FlyBase | FLYB: FBgn0010379 | NA |
| Gene (*D. melanogaster*) | *Tsc1* | FlyBase | FLYB: FBgn0026317 | NA |
| Gene (*D. melanogaster*) | *Tsc2* | FlyBase | FLYB: FBgn0005198 | NA |
| Gene (*D. melanogaster*) | *Pli* | FlyBase | FLYB: FBgn0025574 | NA |
| Gene (*D. melanogaster*) | *sls* | FlyBase | FLYB: FBgn0086906 | NA |
| Gene (*D. melanogaster*) | *geko* | FlyBase | FLYB: FBgn0020300 | NA |
| Gene (*D. melanogaster*) | *Pi3K59F* | FlyBase | FLYB: FBgn0015277 | NA |
| Gene (*D. melanogaster*) | *Pi3K68D* | FlyBase | FLYB: FBgn0015278 | NA |
| Gene (*D. melanogaster*) | *rl* | FlyBase | FLYB: FBgn0003256 | NA |
| Gene (*D. melanogaster*) | *InR* | FlyBase | FLYB: FBgn0283499 | NA |
| Gene (*D. melanogaster*) | *FASN1* | FlyBase | FLYB: FBgn0283427 | NA |
| Gene (*D. melanogaster*) | *FASN3* | FlyBase | FLYB: FBgn0287184 | NA |
| Strain, strain background (*D. melanogaster*) | *w1118* | Vienna *Drosophila* Resource Center | VDRC ID: 60000 | 1w[1118] |
| Genetic reagent (*D. melanogaster*) | *UAS-pvf1i/UAS-pvf1-i1* | Vienna *Drosophila* Resource Center | VDRC ID: 102699 | Genotype: P{KK112211} VIE-260B |
| Genetic reagent (*D. melanogaster*) | *UAS-pvf1-i2* | National Institute of Genetics | NIG Fly: 7103 R-2 | *NA* |
| Genetic reagent (*D. melanogaster*) | *UAS-pvf2-i* | Vienna *Drosophila* Resource Center | VDRC ID: 102072 | Genotype: P{KK110608} VIE-260B |
| Genetic reagent (*D. melanogaster*) | *UAS-pvf3-i* | Vienna *Drosophila* Resource Center | VDRC ID: 105008 | Genotype: P{KK112796} VIE-260B |

*Continued on next page*

*Continued*

| Reagent type (species) or resource | Designation | Source or reference | Identifiers | Additional information |
|---|---|---|---|---|
| Genetic reagent (*D. melanogaster*) | *mus*[ts] | Perrimon Lab stock | PMID:28739899 | Genotype: w[*]; P[w[+mC]=tub]P-Gal80{[ts]}10; P{w[+mC]}=Gal4-{Mef2.R}3 |
| Genetic reagent (*D. melanogaster*) | *v-i* | National Institute of Genetics | NIG Fly: 2115 R-1 | NA |
| Genetic reagent (*D. melanogaster*) | *pli-troj-Gal4* | Bloomington *Drosophila* Stock Center | BDSC: 77693 | Genotype: y[1] w[*]; Mi{Trojan-GAL4.2} Pli[MI00302-TG4.2]/ TM3, Sb[1] Ser[1] |
| Genetic reagent (*D. melanogaster*) | *sls-troj-Gal4* | Bloomington *Drosophila* Stock Center | BDSC: 67495 | Genotype: y[1] w[*]; Mi{Trojan-GAL4.1}sls [MI10783-TG4.1]/ TM3, Sb[1] Ser[1] |
| Genetic reagent (*D. melanogaster*) | *geko-troj-Gal4* | Bloomington *Drosophila* Stock Center | BDSC: 66833 | Genotype: y[1] w[*]; Mi{Trojan-GAL4.1}geko [MI02663-TG4.1]/ TM3, Sb[1] Ser[1] |
| Genetic reagent (*D. melanogaster*) | *pvr*[DN] | Bloomington *Drosophila* Stock Center | BDSC: 58430 | w[1118]; P{w[+mC]=UASp Pvr.DN}D1 |
| Genetic reagent (*D. melanogaster*) | *AT*[ts] | Perrimon lab | FBrf0234453 | yw/w; tub-Gal80ts/ Cyo; Lpp-Gal4/ TM6BTby+ |
| Genetic reagent (*D. melanogaster*) | *oeno*[ts] | Bloomington *Drosophila* Stock Center | BDSC: 65700 | Genotype: P{w[+mC]=Desat1 GAL4.E800}4M, P{w[+mC]=tubP-GAL80[ts]}2 |
| Genetic reagent (*D. melanogaster*) | *pvr-i* | National Institute of Genetics | NIG-FLY: 8222 R-2 | NA |
| Transfected construct | NA | NA | NA | NA |
| Biological sample (*D. melanogaster*) | Adult Abdominal Cuticle | NA | NA | NA |
| Antibody | Anti-Pvr (Rat, Polyclonal) | *Rosin et al., 2004* | PMID:15056618 | 1/400 |
| Antibody | Anti-Pvf1 (Rat, Polyclonal) | *Rosin et al., 2004* | PMID:15056618 | 1/100 |
| Antibody | Anti-p4E-BP (Rabbit, monoclonal) | Cell Signaling | 2855S | 1/200 |
| Recombinant DNA reagent | NA | NA | NA | NA |
| Sequence-based reagent | FASN1 (primer) | FlyPrimerBank | PP32811 | For: ATTATTGAC GCTGGCCTAAACC Rev: TGCTGCTC AGTCTCCGAGT |
| Sequence-based reagent | FASN2 (primer) | FlyPrimerBank | PP1235 | For: ACATTGTGA TCTCGGGACTTTC Rev:CGCTAAAGAA CTTGTCGTCGAA |

*Continued on next page*

*Continued*

| Reagent type (species) or resource | Designation | Source or reference | Identifiers | Additional information |
|---|---|---|---|---|
| Sequence-based reagent | FASN3 (primer) | FlyPrimerBank | PP17912 | For: CGCCGATGG CGTCATTTTAAT Rev:CTCCAAAGAA GGTTGCATCAAAC |
| Sequence-based reagent | Pvf1 (primer) | FlyPrimerBank | PP26981 | For: AATCAACCGTG AGGAATGCAA Rev: GCACGCGGG CATATAGTAGT |
| Sequence-based reagent | ACC (primer) | FlyPrimerBank | PP34306 | For: CGAGCGG GCCATTAGGTTT Rev: GCCATCTTGAT GTATTCGGCAT |
| Peptide, recombinant protein | NA | NA | NA | NA |
| Commercial assay or kit | Chromium Single Cell 3' Library and Gel Bead Kit v2 | 10x Genomics | PN-120267 | NA |
| Commercial assay or kit | Chromium i7 Multiplex Kit | 10x Genomics | PN-120262 | NA |
| Commercial assay or kit | Chromium Single Cell A Chip Kit | 10x Genomics | PN-1000009 | NA |
| Commercial assay or kit | Nuclei Isolation Kit: Nuclei PURE Prep | Sigma-Aldrich | Cat# NUC201-1KT | NA |
| Chemical compound, drug | NA | NA | NA | NA |
| Software, algorithm | Seurat | *Stuart et al., 2019* | PMID:31178118 | NA |
| Software, algorithm | Harmony | *Korsunsky et al., 2019* | PMID:31740819 | NA |
| Software, algorithm | SoupX | *Young and Behjati, 2018* | Biorxiv | doi: https://doi.org/10.1101/303727 |
| Software, algorithm | Biorender | https://biorender.com/ | NA | Biorender was utilized to make the schematic diagrams used in this study. |
| Other | DAPI (nuclear stain) | Vector Laboratories | Cat# H-1200 | Ready to use |
| Other | BODIPY | ThermoFischer | Cat# D3823 | 1/500 dilution of a 1 mg/ml stock in DEPC |
| Other | SyBr Green | Bio-Rad iQ SYBR Green Supermix | Cat# 1708880 | Working concentration: 1X |
| Other | Phalloidin | ThermoFischer | A12381 | 1/100 dilution of a 1 mg/ml stock in Methanol |

## *Drosophila* strains

A detailed list of fly strains and genotypes used for each figure is provided in the Key Resources Table. For tissue-specific transgene expression, we used the following temperature-sensitive strains. *PromE-Gal4, tub-Gal80ts* (BDSC:65406) for the oenocytes (*oeno*ts), *tub-Gal80ts; mef2-Gal4* and *tub-*

*Gal80$^{ts}$; mhc-Gal4* for the muscle (*mus$^{ts}$*), *tub-Gal80$^{ts}$; Lpp-Gal4* for the adipose tissue (*AT$^{ts}$*)and *myo1A-Gal4; tub-Gal80$^{ts}$* for the gut (*gut$^{ts}$*). Control animals throughout the paper were generated by either crossing the temperature-sensitive driver lines to the isogenized *w$^{1118}$* flies from VDRC or by crossing relevant UAS-lines to the isogenized *w$^{1118}$* flies from VDRC (VDRC60000). The *UAS-fasn3* line was generated in the lab. For gene silencing, we used RNAi lines from the TRiP (https://fgr.hms. harvard.edu/fly-in-vivo-rnai) available through BDSC, NIG-Fly (https://shigen.nig.ac.jp/fly/nigfly/) and VDRC (https://stockcenter.vdrc.at/control/main).

## Fly food and temperature

Flies strains were routinely kept at 25°C or 18°C on standard lab food (SF) composed of 15 g yeast, 8.6 g soy flour, 63 g corn flour, 5 g agar, 5 g malt, 74 ml corn syrup per liter. 15 w/v HSD was prepared by adding 10 g of Sucrose to 50 g of standard lab food that was melted using a microwave. The sucrose was thoroughly mixed and dissolved completely before pouring 3 ml of food in standard vials. For all experiments fly crosses were maintained at 18°C on SF. We ensured that the density of our experimental animals was kept optimum for proper growth and uniformity of growth conditions. To do so, all crosses consisted of 8 virgin females that were crossed to six males in standard *Drosophila* food vials. The crosses were allowed to sit in a vial for 6 days at 18°C and then the adults were discarded. We found that 6 days of egg laying at 18°C led to eclosion of 30–40 healthy adult males and an equal number of females. These animals were very comparable in size. We discarded animals that came from vials that yielded $\leq$ 60 adults or $\geq$ 80 adults. We regularly reared multiple crosses in parallel to obtain the final number of males that are necessary for a given experiment. Once adults eclosed, they were aged 3–5 days at 18°C and then shifted to 29°C for respective experimental regimes. For knockdown experiments, flies were maintained at 29°C for 7 days on SF and then shifted to HSD (29°C) for another 7 days before collection. For over-expression experiments, flies were maintained at 29°C for 3–4 days and then transferred to HSD (29°C) for 4–5 days before collection.

For all radioactivity feeding experiments, the radioactive compounds were added in the HSD and feeding was started from day 1 of transfer to HSD (29°C). For lipid mobilization assays, the flies were kept on [1-$^{14}$C]-oleate containing HSD for 3 days and then transferred to cold SF for radioactive chase. For lipid incorporation assays, the flies were transferred to [U-$^{14}$C]-Sucrose containing HSD and samples were collected every 24 hr. U-$^{14}$C-Glucose feeding behavior assay was performed as described before (*Kwon et al., 2015*). For all radio-isotope feeding experiments, we reared the flies on cold food for 7–8 hr between the time that the flies stopped feeding on hot food and the start of our experiment. This ensured that radiolabeled food from the gut is eliminated and that the signal measured comes only from the radioactivity that was incorporated in the cells of the animals (*Kwon et al., 2015*).

## Preparation of radioactive food

4 µCi of radioactive material was prepared in 15 µl Ethanol ([1-$^{14}$C]-oleate) or water ([U-$^{14}$C]-Sucrose) along with 5 µl of FIDC blue food dye (to visually confirm that the radioactive material is spread evenly on the food surface). HSD food was poured into empty vials without creating any bubbles to make sure to have an even surface when the food solidifies. The radioactive mixture was added on the surface of solidified HSD food dropwise using a pipette and making sure that it is evenly distributed. Importantly, the vials were then appended to a rotor that allowed slow rotation of the vials along the longitudinal axis to assist even spreading of the liquid as it dries overnight.

## BODIPY staining and imaging

The abdominal dorsal cuticle was dissected in relaxing buffer (1X PBS, 5 mM MgCl$_2$ and 5 mM EGTA) using micro-scissors with the adipose tissue attached to the cuticle as described before (*Rajan et al., 2017*). The samples were fixed with 4% paraformaldehyde in relaxing buffer for 20 min. Subsequently the samples were washed with PBS, permeabilized with 0.1% PBT (PBS + 0.1% TritonX100) in PBS for 30 min. PBT was removed by washing with PBS, three times, 5 min each, before adding BODIPY. For BODIPY staining, 500 µl of BODIPY in PBS (1/500 dilution of a 1 mg/ml stock in DMSO) was added to the samples and the samples were placed on a rotator for 30 min at room temperature. The samples were then washed in PBS, two times, 10 min each, incubated in PBS

with DAPI for 10 min, washed two more times in PBS, 10 min each, and then mounted with Vecta-shield mounting media (VectaShield 1000). Samples were mounted with a bridge using two pieces of scotch tapes (3M) with the adipose tissue facing the cover slip. Samples were imaged using a Zeiss LSM780 confocal microscope. Images were acquired at room temperature.

## Triacylglycerol (TAG) assays

Whole animal TAG was measured as described previously (*Tennessen et al., 2014*). Briefly, eight males were homogenized in 96-well deep well plates with 250 µl of ice cold PBT using a TissueLyser II (1 mm Zirconium beads, frequency: 30/s, time: 30 s). The plates were centrifuged at 1500 rpm for 3 min to remove any debris and 10 µl of the supernatant and triglyceride standards (Sigma: G7793-5ML) was added to 20 µl of triglyceride reagent (Sigma: T2449-10ML) in 96 well black clear bottom plates (Greiner bio-one; non-binding, black plates 655906) and the mixture was incubated at 37°C for 40–45 min. 100 µl of free-glycerol reagent (Sigma: F6428-40ML) was subsequently added to each well and the samples were incubated at 37°C for 5–10 min before reading absorbance at 540 nm using a 96-well plate reader.

## Immunohistochemistry to detect Pvf1 and PvR proteins

Rat anti-Pvf1 and Rat anti-PvR sera were used at a dilution of 1/200 and 1/500 respectively in blocking solution (BS: 1X PBS+0.1% TritonX100+5% BSA). Anti-Rat secondary antibody was used at a dilution of 1/500. Rabbit monoclonal antibody to p4EBP and secondary antibody (Donkey anti-Rabbit-488) for p4EBP detection were pre-cleared with fixed embryos ($w^{1118}$) at a dilution of 1/50 in BS and then used at a dilution of 1/200 and 1/500 in BS, respectively.

### Oenocytes/adipose tissue

The abdominal dorsal cuticle was dissected in relaxing buffer (1X PBS, 5 mM $MgCl_2$ and 5 mM EGTA) using micro-scissors with the adipose tissue attached to the cuticle. The samples were fixed with 4% paraformaldehyde in relaxing buffer for 20 min. The samples were then washed in 1X PBS three times, 5 min each. The samples were permeabilized with 1X PBS+0.1% TritonX100 (PBT) for 30 min and then incubated with BS for 2 hr at room temperature. Primary antibody staining was performed in BS at 4°C overnight or 48 hr for staining the oenocytes with constant rotation (a long incubation is necessary for the antibody to percolate into the oenocytes) with constant rotation. Post primary incubation, samples were washed generously (five times, 15 min washes with PBT) at room temperature. Secondary, antibody incubation was done at room temperature for 2 hr in BS diluted 1/5 in PBT. Post secondary-incubation samples were washed in PBT and mounted with Vectashield mounting media. For phalloidin staining, PBT was exchanged with PBS with three washes and samples were incubated with phalloidin in PBS for 30 min at RT. Samples were then washed in PBS and mounted with vecta-shield mounting media.

### Muscle

Adult male thoraxes were prepared for fixation by removing the head and abdomen. Additionally, the tips of the legs were cut using micro-scissors to allow easy access to the leg muscles for the fixative. The dissected samples were fixed in 4% paraformaldehyde for 30 min. The samples were subsequently embedded in 4% low-melt agarose and left at 4°C over night for the agarose to solidify. Samples in agarose blocks were mounted on to the stage of a vibratome (Leica VT1000M) in ice-cold PBS and 100 µm sections were cut. The sections were further trimmed under the microscope tissue sections with some surrounding agarose were transferred to 2 ml tubes. Subsequent immunostaining steps were similar to as discussed in the previous section. Stained sections were mounted in Vectashield mounting media and imaged using a Zeiss LSM 780 confocal microscope.

## RT-qPCR

Real-time quantitative PCR (RT-qPCR) experiments were conducted using a Biorad CFX 96/384 device. The iQ SYBR green super mix and i-Script RT-reaction mix was used as per the manufacturer's instructions for the qPCR reactions. 7.5 µl of the 2x reaction mix, primer-mix final concentration of 133 nM and 20 ng of cDNA (assuming 1:1 conversion of RNA to cDNA) were regularly used per reaction. Reagents and samples were dispensed using a Formulatrix Mantis small volume liquid

handler. The ΔΔCt method was used to calculate fold change in experimental conditions. Four to five biological replicates were used in all experiments. Transcript levels were normalized to *Drosophila Rp49*, *tub* and *gapdh*. Standard curves were run for each primer before use and we only used primers that showed an efficiency above 95%.

## Folch extraction and thin-layer chromatography (TLC)

### Folch extraction

Animals were collected in 1.5 ml screw cap tubes along with 1 mm zirconium beads and were frozen on dry ice. Samples were stored at −80˚C if necessary, until all samples were ready for processing. For processing of the samples, 600 µl of Methanol:ChCl$_3$:H$_2$O (10:5:4) was added to the tubes using a graduated glass Hamilton syringe and the animals were homogenized using the Qiagen Tissue-Lyser II instrument. Two rounds of homogenization at a frequency of 30/sec and total duration of 30 s per round were performed to ensure complete homogenization of the tissues. The samples were placed on a rotor at 37˚C for 1 hr. Subsequently, 160 µl of ChCl$_3$ and 160 µl of 1 M KCl were added to each sample. The samples were shaken vigorously for 5 s and briefly vortexed before centrifuging at 3000 rpm for 2 min on a standard Eppendorf bench top centrifuge for phase separation. 220 µl of the organic phase (bottom layer) was pipetted out into new clean 1.5 ml centrifuge tubes using a graduated glass Hamilton syringe. The samples were then placed in a vacuum concentrator (Labcono centrivap console) and the organic solvent was completely removed. The dried samples were stored at −80 if needed or run directly on a TLC plate.

### Thin-layer chromatography

For TLC, the dried samples were resuspended in 40 µl CHCl$_3$ and the entire volume was loaded on to Analtech channeled TLC plates with preadsorbent zones that allow loading of large volumes of samples (Analtech P43911). The lipids were then separated using hexane:diethyl ether:acetic acid (80:20:1) solvent system. The plates were exposed to phosphor imager screens overnight and revealed by using a Typhoon FLA 7000 phosphor imager. The density of the TAG bands was calculated using imageJ/Fiji.

## Preparation of single nuclei suspension from adult *Drosophila* abdominal cuticle

Forty adult male abdomens were quickly dissected in ice cold relaxing buffer (1xPBS, 10 mM EGTA and 10 mM MgCl$_2$) by cutting off the last 2nd and abdominal segment with micro-scissors and hollowing out the abdomen by removing the intestines and male reproductive organs. The dissected tissue was then roughly chopped and transferred to a Dounce homogenizer. The relaxing buffer was replaced with 1.3 ml of nuclei lysis buffer from the Sigma NUC-201 Nuclei isolation kit. Single nuclei suspension from the homogenate was prepared according to the manufacturer's instruction the in the NUC-201 kit. Briefly, a 1.5 M sucrose cushion was used for generating the gradient for density gradient centrifugation. Density gradient centrifugation was performed in 2 ml tubes using a SW55Ti rotor on a Beckman ultracentrifuge.

## Analysis of snRNA-Seq data

We used the 10X Genomics cellranger count pipeline (version 3.1.0) to analyze the demultiplexed FASTQ data and generated the single cell count matrix, once for each sample. We aligned the reads to a custom 'pre-mRNA' reference which was generated as described by 10X Genomics based on FlyBase R6.29. We applied SoupX (version 0.3.1) (*Young and Behjati, 2018*) to directly correct the count matrix from cellranger with fixed contamination value equals 0.45 for each sample. We filtered the cells beyond UMI counts ± 2 fold Standard Deviation of the average total sample counts (log10) after SoupX, which were regarded as doublets or dead cells in droplet. The quality filtered datasets were combined into a single Seurat (version 3.1.2) object (*Stuart et al., 2019*) and integrated using Harmony (version 1.0) (*Korsunsky et al., 2019*) with default analysis workflow and parameters. FindClusters function in Seurat was applied to identify cell clusters. The number of cell clusters identified depended on the number of PCs used for the clustering analysis (resolution). A resolution of 0.1 was chosen as clustering parameter for *Figure 2B*. A resolution of 0.4 was chosen as clustering parameter for *Figure 2—figure supplement 3A*. The code for the snRNA-Seq analysis can be found

at https://github.com/liuyifang/Drosophila-PDGF-VEGF-signaling-from-muscles-to-hepatocyte-like-cells-protects-against-obesity (*Ghosh, 2020*; copy archived at; swh:1:rev:f1ad799015c901dad378-f6e488dc38f4a19fd703) Dot-and-Violin plots were generated using the Seurat DotPlot and VlnPlot functions. We performed pathway enrichment analysis on marker genes with positive fold change for each cluster as described in *Tang et al., 2018*. Gene sets of Transcription Factor (TF) target genes of major signaling pathways were assembled manually (unpublished data). Enrichment p-value was calculated based on the hypergeometric distribution using the background of 11863 genes identified as expressed in this dataset. The strength of enrichment was calculated as negative of log10(p-value), which is used to plot the heatmap. The snRNA-seq data is available at the gene expression omnibus (GEO) under the accession code GSE147601.

## Quantification and statistical analysis

Graphical representation and statistical analysis of all quantitative data was performed using GraphPad Prism eight software (www.graphpad.com). Quantification of lipid droplet size was performed using CellProfiler and the pipeline used will be made available upon request to the corresponding author (*Lamprecht et al., 2007*). For each data point a total of about 10 images acquired from one of the adipose tissue lobes of each animal was analyzed. We counted hundreds of lipid droplets per image and the mean lipid droplet size from these images is reported as a single data point per animal. Quantification of fluorescent intensities of immunostained samples was performed using a custom-made ImageJ (Fiji) macro (also available upon request).

For starvation resistance studies, the data is presented as Kaplan-Meier survival plots. Statistical significance for the survival plots was determined using the Log-Rank (Mantel-Cox) test. For all other quantifications, including the data for TAG quantification, the data is presented as either bar plots or dot plots with the error bars showing standard error of mean. Methodologies used for determining statistical significance are mentioned in the figure legends. Asterisks illustrate statistically significant differences between conditions. $****p<0.0001$, $0.0001<***p<0.001$, $0.001<**p<0.01$ and $0.01<*p<0.05$.

## Acknowledgements

The authors thank the Bloomington *Drosophila* Stock Center (BDSC), Vienna *Drosophila* Resource Center (VDRC), Fly stocks of National Institute of Genetics (NIG-Fly) and the Transgenic RNAi project (TRiP) for fly stocks; the Microscopy Resources on the North Quad (MicRoN) for access to their laser scanning confocal microscopes; the Biorender team (https://biorender.com) for help with making illustrations for *Figure 6F*; Dr. Chandramohan Chitraju for his expert advice on designing and running the TLC experiments; Dr. Tobias C Walther and Dr. Robert V Farese for allowing us to use their Typhoon scanner and Dr. Stephanie S Mohr, Dr. Patrick Jouandin and Dr. Ben Ewen-Campen for their comments on the manuscript. VB and SHS were supported by funds from the Harvard Medical School Tools and Technology Committee. ACG was supported by a postdoctoral fellowship from the American Heart Association (18POST33990414). This work was supported by the National Institute of Health (5RO1AR05735210 and P01CA120964). NP is an investigator of the Howard Hughes Medical Institute.

## Additional information

### Funding

| Funder | Grant reference number | Author |
| --- | --- | --- |
| American Heart Association | 18POST33990414 | Arpan C Ghosh |
| National Institutes of Health | 5RO1AR05735210 | Norbert Perrimon |
| National Institutes of Health | P01CA120964 | Norbert Perrimon |
| Howard Hughes Medical Institute | | Norbert Perrimon |
| Harvard Medical School Tools and Technology Committee | | Victor Barrera Shannan J Ho Sui |

The funders had no role in study design, data collection and interpretation, or the decision to submit the work for publication.

## Author contributions
Arpan C Ghosh, Conceptualization, Formal analysis, Supervision, Funding acquisition, Validation, Investigation, Visualization, Methodology, Writing - original draft, Project administration; Sudhir Gopal Tattikota, Formal analysis, Investigation, Visualization, Writing - original draft, Writing - review and editing; Yifang Liu, Data curation, Software, Formal analysis, Visualization, Methodology, Writing - review and editing; Aram Comjean, Data curation, Software, Visualization; Yanhui Hu, Data curation, Software, Visualization, Writing - review and editing; Victor Barrera, Shannan J Ho Sui, Software, Writing - review and editing; Norbert Perrimon, Conceptualization, Resources, Supervision, Funding acquisition, Investigation, Project administration, Writing - review and editing

## Author ORCIDs
Arpan C Ghosh (iD) https://orcid.org/0000-0001-6553-938X
Sudhir Gopal Tattikota (iD) https://orcid.org/0000-0003-0318-5533
Victor Barrera (iD) http://orcid.org/0000-0003-0590-4634

## Decision letter and Author response
Decision letter https://doi.org/10.7554/eLife.56969.sa1
Author response https://doi.org/10.7554/eLife.56969.sa2

# Additional files
## Supplementary files
• Supplementary file 1. Table representing number of nuclei, genes and unique molecular identifiers (UMIs) recovered per nuclei per sample.

• Supplementary file 2. Marker genes identified in each of the 10 clusters (*Figure 2B*) identified in our snRNA seq of control (*oeno^{ts}>EGFP*) animals.

• Supplementary file 3. Signaling pathway genes identified in our snRNA sequencing analysis and used for pathway enrichment analysis (*Figure 2C*).

• Supplementary file 4. Normalized expression levels (obtained from snRNA seq analysis) of starvation responsive genes in the oenocytes of control (*oeno^{ts}>EGFP*) animals and animals lacking TOR signaling in the oenocytes (*oeno^{ts}>tsc1, tsc2*).

• Transparent reporting form

## Data availability
Sequencing data have been deposited in GEO under the accession number GSE147601. Elsewhere, data can be visualized at: www.flyrnai.org/scRNA/abdomen/. Data code can accessed at: https://github.com/liuyifang/*Drosophila*-PDGF-VEGF-signaling-from-muscles-to-hepatocyte-like-cells-protects-against-obesity (copy archived at https://archive.softwareheritage.org/swh:1:rev:f1ad799015c901dad378f6e488dc38f4a19fd703/).

The following datasets were generated:

| Author(s) | Year | Dataset title | Dataset URL | Database and Identifier |
|---|---|---|---|---|
| Ghosh AC, Tattikota SG, Yifang L, Comjean A, Yanhui H, Barrera V, Sui SJ, Perrimon N | 2020 | *Drosophila* PDGF/VEGF signaling from muscles to hepatocyte-like cells protects against obesity | https://www.ncbi.nlm.nih.gov/geo/query/acc.cgi?acc=GSE147601 | NCBI Gene Expression Omnibus, GSE147601 |
| Ghosh AC, Tattikota SG, Yifang L, Comjean A, Yanhui H, Barrera V, | 2020 | *Drosophila* PDGF/VEGF signaling from muscles to hepatocyte-like cells protects against obesity | https://www.flyrnai.org/scRNA/abdomen/ | DRSC/TRiP Functional Genomics Resources, abdomen |

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
