## [Decision Letter]

Thank you for submitting your article "*Drosophila* PDGF/VEGF signaling from muscles to hepatocyte-like cells protects against obesity" for consideration by *eLife*. Your article has been reviewed by three peer reviewers and the evaluation has been overseen by Jonathan Cooper as the Senior Editor. The reviewers have opted to remain anonymous.

The reviewers have discussed the reviews with one another and the Reviewing Editor has drafted this decision to help you prepare a revised submission.

As the editors have judged that your manuscript is of interest, but additional experiments are required before it is published, we would like to draw your attention to changes in our revision policy that we have made in response to COVID-19 (https://elifesciences.org/articles/57162). First, because many researchers have temporarily lost access to their labs, we will give authors as much time as they need to submit revised manuscripts. We are also offering, if you choose, to post the manuscript to bioRxiv (if it is not already there) along with this decision letter and a formal designation that the manuscript is "in revision at *eLife*". Please let us know if you would like to pursue this option. (If your work is more suitable for medRxiv, you will need to post the preprint yourself, as the mechanisms for us to do so are still in development.)

Summary:

This manuscript by Ghosh et al. describes a role for the *Drosophila* PDGF/VEGF homolog *pvf1* in regulating adult lipid accumulation. Using a series of tissue-specific RNAi experiments, the authors demonstrate that knockdown of *pvf1* expression in adult muscles induces excess lipid accumulation in adipose tissues and oenocytes. This result suggests that *pvf1* acts as a myokine that inhibits lipid accumulation in these tissues. Consistent with this possibility, inhibition of PvR activity in the oenocytes, but not the fat body, also led to elevated lipid accumulation, indicating that *pvf1* regulates systemic lipid metabolism by acting through these hepatocyte-like cells. These results, together with a series of ^14^C pulse-chase experiments, are quite important because they reveal that lipid synthesis in adults is controlled by a muscle-derived signal acting on the oenocytes.

After careful review and discussion all three reviewers have agreed that this is exciting work with a few major points requiring attention.

Essential revisions:

1) Oenocytes are well known to accumulate lipid droplets upon starvation. Since Tor functions as a key nutrient sensor, any disruption of Tor activity within these cells could trigger a starvation response and induce lipid accumulation. Based on earlier studies (see for example Chatterjee et al.), the findings in this manuscript could easily be reinterpreted, that loss of Pvf1 signaling triggers the oenocytes to inappropriately activate a starvation response. Could a starvation response be the basis of what is going on here? Please address.

2) The authors state that Tor activity within the oenocytes is independent of insulin signaling. The reviewers see little evidence for this statement based on what is known about the unusual way that IIS signaling influences lipid metabolism in oenocytes. As demonstrated by Chatterjee et al., expression of an active InR is required for lipid droplet accumulation in oenocytes. The experiment described in Figure 3D,E of this manuscript gave exactly the expected result and one that was previously shown by Chatterjee et al. overexpression of InR dominant negative does not induce excess lipid accumulation in oenocytes (this previous result should be referenced and adequately discussed in the manuscript). If the authors want to demonstrate that the PvR-RNAi phenotype within the oenocytes is independent of IIS signaling regarding lipid accumulation, they should coexpress the InR-dominant negative construct with PvR-RNAi in the oenocytes and look for suppression of the excess lipid phenotype. Also, what happens to lipid droplet accumulation and total TAGs if constitutive active InR is overexpressed? And Tcs1-2 are downregulated? Please address experimentally.

3) Growth and metabolism are tightly coregulated. Since most of the genes manipulated here also control cellular growth, is there a correlation between differences in the size of oenocytes and how much lipid content they accumulate? How can the authors distinguish their model from a model that suggests that Pvf1/PvR expression at the proper timing in muscle/oenocyte is required for the proper development of proper sized oenocytes? And that lack of such proper development would impair their function? Please address this experimentally and in the Discussion.

4) In general the model where PVF1 plays a role in re-establishing lipid metabolism in adults is appealing. Reviewers request the authors discuss alternative interpretations of their results and potential alternative models, as the data presented here can support various models. Related to this, in the Discussion please develop the paragraph describing adult fat body development. It is oversimplified and seems to suggest that the same larval fat body expands to make an adult fat body. From what has been (still poorly) studied these two tissues have different origins (see for example Aguila et al., 2007 and Nelliott, Bond and Hoshizaki, 2006).

Additionally, the authors claim that newly emerged flies contain lower lipid stores that must be restored in the adult. The situation is a bit more complicated, as shown by Aguila et al. (2007): larval fat body survives in newly emerged flies as floating cells, making these flies more resistant to fasting. Consistently, Wicker-Thomas et al. (2015) and Storelli et al. (2019) have shown that total amounts of TAGs are higher in newly emerged than in older fed flies. The word "restoration" suggests that the adult fat body needs to be replenished, but it is not known and rather unlikely that adult fat body contains high lipid stores in pupae. Prepupae contain high lipid stores that are used to some extent during metamorphosis and newly emerged flies still contain high lipid content in the remaining larval fat body (Storelli et al., 2018). Adult feeding does not result in restoration of lipid stores, rather lipids incorporate into the adult fat body, and the overall TAG levels never reaches those of newly emerged flies. Please rewrite the conclusions to provide a clearer picture of how your newly proposed model fits with what is known about fat body restructuring during the transition into adulthood.

5) The data must be revised with additional controls, and the number of animals per sample, and number of biological repeats, should be clarified. Genetics, growth conditions and age strongly influence metabolic and lifespan outcomes. Therefore, we request that the following controls are presented for the experiments:

a) All experiments need the genetic background control for each line, not just the drivers but also the RNAi lines and UAS-overexpression lines (unless all lines have been backcrossed and isogenized to the same background).

b) Each experiment needs an overexpression control (GFP or similar) or a RNAi control (as done in Figure 1C).

c) Careful control of growth conditions is required in metabolic research. Overcrowding or underpopulation of animals per vial/bottle can influence their ultimate size and lipid amount. How were these variables controlled and what were the conditions?

d) Changes in temperature also greatly alter metabolic outcomes in flies. What controls were done for temperature changes? Temperature shifts have to be properly controlled and described for all experiments.

[Editors' note: further revisions were suggested prior to acceptance, as described below.]

Thank you for re-submitting your article "*Drosophila* PDGF/VEGF signaling from muscles to hepatocyte-like cells protects against obesity" for consideration by *eLife*. Your article has been reviewed by three peer reviewers, one of whom is a member of our Board of Reviewing Editors, and the evaluation has been overseen by Jonathan Cooper as the Senior Editor. The reviewers have opted to remain anonymous.

In this study, Ghosh and collaborators investigate the physiological role of muscle Pvf1, the *Drosophila* homologue of PDGF/VEGF, in organismal metabolism. Currently, homeostatic mechanisms that regulate energy homeostasis are still poorly known and explored. The whole organismal context of these studies, involving physiology and several organs, make these studies exciting and powerful.

In the revised version, Ghosh and collaborators provide novel data that clearly dissociate the described phenotype from that induced by starvation. In their resubmission the authors have addressed most of our major concerns. However, we request that the authors address three remaining major points:

Major points:

1) We maintain our initial reservation about the fact that the authors do not show in Figure 5A that "these flies are not defective in production of VLCFAs needed for waterproofing of the cuticle". We request this point be removed.

2) We also request the authors reproduce in the body text of the Results (not the Materials and methods) their statement of justification for why overexpression and RNAi controls were not used (response to point 5c). I suspect that many readers will share our concerns about the authors' approach and I think it is crucial that readers are able to decide for themselves whether there might be caveats to the authors' conclusions.

3) It is widely known that crosses from different genotypes, and especially small crosses as used by the authors (8 virgins with 6 males), often result in a wide range of numbers of healthy progeny. It is for this reason that, in order to tightly control for growth conditions, crosses are typically performed with a higher number of parents, and an equal number of progeny (often as L1s) are transferred from the vial to new growth media. Since this was not the approach, we request the authors be more precise when they state that the range of numbers of healthy adults obtained from each cross was "about 80". If the range was, for example, 75-85, then this is acceptable; if the range was 50-110, then this is a point of concern. We think this is crucial information that can support readers with the conclusions they make when interpreting the data.

---

## [Author Response]

Essential revisions:1) Oenocytes are well known to accumulate lipid droplets upon starvation. Since Tor functions as a key nutrient sensor, any disruption of Tor activity within these cells could trigger a starvation response and induce lipid accumulation. Based on earlier studies (see for example Chatterjee et al.), the findings in this manuscript could easily be reinterpreted, that loss of Pvf1 signaling triggers the oenocytes to inappropriately activate a starvation response. Could a starvation response be the basis of what is going on here? Please address.

The reviewers have proposed an alternate hypothesis that can explain why animals lacking PvR/TOR signaling in the oenocytes accumulate lipids in the oenocytes. We agree that a starvation like response to loss of TOR in the oenocyte can lead to lipid accumulation in the oenocytes. We first tested whether flies lacking the muscle to oenocyte Pvf1 signaling axis have a systemic starvation-like response. We measured the expression levels of bmm, thor and InR (genes that are upregulated in response to starvation) in the Adipose tissue of animals lacking Pvf1 in their muscle or PvR signaling in their oenocytes. Compared to control animals, both experimental animals did not show any change in the expression of these genes indicating a lack of systemic starvation response.

We next tested the possibility that loss of PvR/TOR pathway in the oenocyte leads to inappropriate activation of the starvation response in the oenocytes. As reported in our original version of the paper, we performed single-nuclei RNA sequencing of wild type adult abdominal cuticles and were able to identify the oenocyte-specific gene expression using this technique. In addition, we performed a similar sn-RNAseq of nuclei isolated from the abdominal cuticle of *oeno>TOR^KD^* (*oeno>tsc1, tsc2*) flies. To identify differentially expressed genes, we converted the oenocyte-specific gene expression profiles from both data sets to pseudobulk expression for the genes that were detected. This allowed us to compare the expression profiles of the oenocytes from control animals and animals lacking TOR signaling in the oenocytes. Next, we specifically looked at the effect of loss of TOR on the expression of the 47 genes that Chaterjee et al. had reported to be up-regulated specifically in the oenocytes in response to starvation. Interestingly, we did not observe any significant change in the expression of any of these genes in our experimental flies. These observations support that loss of TOR does not lead to a starvation like response in the oenocytes. We have included the new dataset and discussed the proposed hypothesis in the revised manuscript.

2) The authors state that Tor activity within the oenocytes is independent of insulin signaling. The reviewers see little evidence for this statement based on what is known about the unusual way that IIS signaling influences lipid metabolism in oenocytes. As demonstrated by Chatterjee et al., expression of an active InR is required for lipid droplet accumulation in oenocytes. The experiment described in Figure 3D,E of this manuscript gave exactly the expected result and one that was previously shown by Chatterjee et al. overexpression of InR dominant negative does not induce excess lipid accumulation in oenocytes (this previous result should be referenced and adequately discussed in the manuscript). If the authors want to demonstrate that the PvR-RNAi phenotype within the oenocytes is independent of IIS signaling regarding lipid accumulation, they should coexpress the InR-dominant negative construct with PvR-RNAi in the oenocytes and look for suppression of the excess lipid phenotype. Also, what happens to lipid droplet accumulation and total TAGs if constitutive active InR is overexpressed? And Tcs1-2 are downregulated? Please address experimentally.

We provide a number of evidence that shows that under steady state conditions TOR signaling in the oenocytes suppresses lipid accumulation in the oenocytes and in the adipose tissue.

First, we show that oenocyte-specific loss of TOR signaling by over-expression of *tsc1* and *tsc2* leads to massive accumulation of lipids in the oenocytes and the adipose tissue (Figure 3B,C).

Second, we show that oenocyte-specific activation of TOR by knocking down *tsc2* can completely rescue the lipid accumulation phenotype observed in the *oeno^ts^>PvR^DN^* flies (Figure 4D, E).

Third, we show that oenocyte-specific activation of TOR by over-expressing a constitutively active form of Rheb can also completely rescue the lipid accumulation of phenotype observed in the *oeno^ts^>PvR^DN^* flies (Figure 4—figure supplement 1).

Assuming that InR in the oenocyte is involved in the activation of the TOR pathway under steady state feeding conditions, it is reasonable to predict that loss of InR signaling should mimic the phenotype of loss of TOR signaling. However, our data shows that this is not the case and that loss of InR signaling does not lead to excessive accumulation of lipids (Figure 3D). Additionally, we show that loss of InR signaling in the oenocytes of feeding animals does not have any effect on p4EBP levels, further supporting the claim that in feeding steady state animals InR does not activate TOR (Figure 4B,C). Moreover, we find that loss of InR signaling reduces oenocyte size whereas loss of TOR does not. These discrepancies between the phenotypes caused by loss of InR signaling and loss of TOR signaling in the oenocytes strongly suggest that InR does not activate TOR pathway in the oenocytes of feeding *Drosophila* adults. However, since our data does not comprehensively investigate whether InR signaling works independently to TOR in the oenocyte, we have stopped short of making that claim and only present the data as it is.

In our original work, we had not tested whether oenocyte-specific activation of TOR can lead to lipid accumulation in the oenocyte or adipose tissue. Neither had we tested whether InR signaling is still necessary for the manifestation of the lipid accumulation phenotypes we see in our TOR knockdown animals. We thank the reviewers for suggesting experiments that directly test these possibilities which we have now included.

Specifically, in our revised manuscript we present data that shows that oenocyte specific activation of TOR signaling (via oenocyte specific knock down of *tsc2*) does not lead to accumulation of lipids either in the oenocyte or the adipose tissue (Figure 4—figure supplement 1C,D). While this does not challenge the findings of the Chatterjee et al. paper that show that InR signaling is required for oenocyte lipid build up, this data does show that InR signaling most likely works independent of TOR in mediating its effects on lipid accumulation.

We also checked the effect of co-expressing *pvr^DN^* and *inr^DN^* in the oenocyte on lipid accumulation. Oenocyte specific loss of InR signaling did not suppress the accumulation of lipids in adipose tissue observed in flies co-expressing *pvr^DN^* in the oenocytes (Figure 4—figure supplement 2A,B). However, we were surprised to see that co-suppression of InR signaling did rescue the accumulation of lipid in the oenocytes. This data shows that InR signaling is indeed required for accumulation of lipids in the oenocyte but the role does not extend to the systemic effects on lipid accumulation that oenocyte specific loss of PvR signaling has.

Understanding the complex interplay between InR, PvR and TOR in feeding and starvation conditions will be an exciting direction to pursue in the future.

3) Growth and metabolism are tightly coregulated. Since most of the genes manipulated here also control cellular growth, is there a correlation between differences in the size of oenocytes and how much lipid content they accumulate? How can the authors distinguish their model from a model that suggests that Pvf1/PvR expression at the proper timing in muscle/oenocyte is required for the proper development of proper sized oenocytes? And that lack of such proper development would impair their function? Please address this experimentally and in the Discussion.

The reviewer raises a very valid point. It is indeed possible that changes in the size of oenocytes can affect lipid metabolism. Since all of our studies involved conditional genetic manipulations in the oenocytes on post developmental adult flies that were at least 5 days old, it is quite unlikely that our experimental animals have developmental defects in the oenocytes. It is however possible that loss of some of these pathways could influence post developmental size of the oenocytes by affecting the size of the cells. We do observe that loss of either InR signaling or Akt1 in the oenocytes causes a strong reduction in overall size of the oenocytes. Contrarily, loss of TOR signaling tends to cause an overall increase in oenocyte size, although this change was not significant when measured. Loss of PvR signaling did not have any obvious effects on oenocyte size. Interestingly, we did not observe any correlation between the size of the oenocytes and the obesity phenotype. Based on this observation and the observation that loss of PvR signaling in the oenocyte did not cause any change in oenocyte size we conclude that the muscle-to-oenocyte Pvf1 signaling axis does not regulate lipid accumulation by affecting oenocyte size. We have presented the relevant data in the updated manuscript (Figure 4—figure supplement 2C,D) and have discussed the above mentioned possibilities in the Discussion.

4) In general the model where PVF1 plays a role in reestablishing lipid metabolism in adults is appealing. Reviewers request the authors discuss alternative interpretations of their results and potential alternative models, as the data presented here can support various models. Related to this, in the Discussion please develop the paragraph describing adult fat body development. It is oversimplified and seems to suggest that the same larval fat body expands to make an adult fat body. From what has been (still poorly) studied these two tissues have different origins (see for example Aguila et al., 2007 and Nelliott, Bond and Hoshizaki, 2006).Additionally, the authors claim that newly emerged flies contain lower lipid stores that must be restored in the adult. The situation is a bit more complicated, as shown by Aguila et al. (2007): larval fat body survives in newly emerged flies as floating cells, making these flies more resistant to fasting. Consistently, Wicker-Thomas et al. (2015) and Storelli et al. (2019) have shown that total amounts of TAGs are higher in newly emerged than in older fed flies. The word "restoration" suggests that the adult fat body needs to be replenished, but it is not known and rather unlikely that adult fat body contains high lipid stores in pupae. Prepupae contain high lipid stores that are used to some extent during metamorphosis and newly emerged flies still contain high lipid content in the remaining larval fat body (Storelli et al., 2018). Adult feeding does not result in restoration of lipid stores, rather lipids incorporate into the adult fat body, and the overall TAG levels never reaches those of newly emerged flies. Please rewrite the conclusions to provide a clearer picture of how your newly proposed model fits with what is known about fat body restructuring during the transition into adulthood.

We agree that the information on adult adipose tissue development was not articulated clearly and could be mis-interpreted. We have now rephrased the section in the Discussion and also changed the wordings in the Abstract to better reflect the sequence of events that leads to adult adipose tissue development and lipid build up in the adipose tissue of a young adult.

5) The data must be revised with additional controls, and the number of animals per sample, and number of biological repeats, should be clarified. Genetics, growth conditions and age strongly influence metabolic and lifespan outcomes. Therefore, we request that the following controls are presented for the experiments:a) All experiments need the genetic background control for each line, not just the drivers but also the RNAi lines and UAS-overexpression lines (unless all lines have been backcrossed and isogenized to the same background).

The adult adipose tissue dissection, staining and imaging protocol is quite laborious and makes inclusion of a large number of controls in every experiment quite difficult. As such we had kept the number of control genotypes in a given experiment low to have a manageable number of genotypes. Further, although some of our experiments could have been performed with additional control lines, we believe that the fact that we used more than one genetic approach to drive home our conclusions and that those independent approaches culminated to the same conclusion addresses the issue.

At this point it is impossible for us to repeat all our experiments with additional controls. However, since all our imaging was done under identical conditions it is possible to compare lipid droplet quantifications between experiments. We would like to bring to light that the 5 genetic background controls used in the original manuscript

pvf1-i/+

oeno^ts^/+

oeno^ts^>UAS

inr^DN^/+

tsc2-i/+

all show very similar average adipose tissue lipid droplet size. We have now analyzed and compared the lipid droplet size of 4 additional controls lines (Figure 4—figure supplement 3)

akt1-i/+

Pi3K21B-i/+

Pi3K92E-i/+

UAS-tsc1,tsc2/+

to the *oeno^ts^* line that we have most commonly used as a control. The new data demonstrates that all our controls show similar baseline adipose tissue lipid droplet size across multiple genotypes.

b) Each experiment needs an overexpression control (GFP or similar) or a RNAi control (as done in Figure 1C).

While GFP overexpression or a control RNAi expression is often used in the field as valid controls, they by no means represent ideal controls for our experiments and each of them come with significant drawbacks. Overexpressing GFP in *Drosophila* cells could be toxic in some contexts leading to unpredictable physiological effects. Similarly knocking down another unrelated gene such as Vermillion could also have unpredictable effects on the biology being studied.

However, we think that the best way to test a hypothesis is by making sure that multiple lines of evidence converge at the same conclusion and not necessarily by playing with a large number of control lines. As explained in the previous point, we do draw our conclusions based on multiple experimental evidences that point toward the same logical explanation.

c) Careful control of growth conditions is required in metabolic research. Overcrowding or underpopulation of animals per vial/bottle can influence their ultimate size and lipid amount. How were these variables controlled and what were the conditions?

The growth conditions and densities of animals were tightly controlled to make sure that they were not the deciding factor for our observations. The details involved are now included in the updated Materials and methods section

“We ensured that the density of our experimental animals was optimal for proper growth and uniformity of growth conditions. To do so, all crosses consisted of 8 virgin females that were crossed to 6 males in standard *Drosophila* food vials. The crosses were allowed to remain in a vial for 6 days at 18 ^o^C and then the adults were discarded. We found that 6 days of egg laying at 18 ^o^C led to eclosion of about 80 healthy adults that were very comparable in size. We often reared multiple crosses in parallel to obtain the final number of males that are necessary for a given experiment.”

d) Changes in temperature also greatly alter metabolic outcomes in flies. What controls were done for temperature changes? Temperature shifts have to be properly controlled and described for all experiments.

We have controlled our temperature shifts meticulously and the protocol is described in the Materials and methods section. Indeed, temperature shifts could have effects on metabolism, however since both our control and experimental flies were temperature shifted at the same time and were maintained at 29°C for at least 6 days before any experiments, we assume that any effects of temperature equally influenced all genotypes and any effects that we see in our experimental flies come from the genetic manipulations we are doing. Thus, we stand by our conclusions, which are well controlled at the level of rearing our control and experimental flies.

[Editors' note: further revisions were suggested prior to acceptance, as described below.]

Major points:1) We maintain our initial reservation about the fact that the authors do not show in Figure 5A that "these flies are not defective in production of VLCFAs needed for waterproofing of the cuticle". We request this point be removed.

We have removed the above-mentioned line from the text as requested.

2) We also request the authors reproduce in the body text of the Results (not the Materials and methods) their statement of justification for why overexpression and RNAi controls were not used (response to point 5c). I suspect that many readers will share our concerns about the authors' approach and I think it is crucial that readers are able to decide for themselves whether there might be caveats to the authors' conclusions.

We have moved our response to both point 5b and 5c of the original decision letter to the main body of the paper in the Results section as requested.

3) It is widely known that crosses from different genotypes, and especially small crosses as used by the authors (8 virgins with 6 males), often result in a wide range of numbers of healthy progeny. It is for this reason that, in order to tightly control for growth conditions, crosses are typically performed with a higher number of parents, and an equal number of progeny (often as L1s) are transferred from the vial to new growth media. Since this was not the approach, we request the authors be more precise when they state that the range of numbers of healthy adults obtained from each cross was "about 80". If the range was, for example, 75-85, then this is acceptable; if the range was 50-110, then this is a point of concern. We think this is crucial information that can support readers with the conclusions they make when interpreting the data.

We regularly retrieved ~30-40 males and an equal number of females from our crosses that were carried out in standard 10 cm food vials. We discarded vials that had an aberrantly low (≤60) or high number (≥80) of adults eclosing from them. We have now added this information in the text.